# The *Shigella flexneri* effector IpaH1.4 facilitates RNF213 degradation and protects cytosolic bacteria against interferon-induced ubiquitylation

Luz Saavedra-Sanchez[1], Mary S Dickinson[1], Shruti S Apte[1], Yifeng Zhang[1], Maarten De Jong[2], Samantha Skavicus[1], Nicholas S Heaton[1], Neal M Alto[2], Jorn Coers[1,3]*

[1]Department of Molecular Genetics and Microbiology, Duke University Medical Center, Durham, United States; [2]Department of Microbiology, University of Texas Southwestern Medical Center, Dallas, United States; [3]Department of Integrative Immunobiology, Duke University Medical Center, Durham, United States

## eLife Assessment

In this manuscript, the authors report the **fundamental** finding that a secreted ubiquitin ligase of Shigella, called IpaH1.4, mediates the degradation of a host defense factor, RNF213. The data are **convincing** and represent a major contribution to our understanding of cell-autonomous immunity and bacterial pathogenesis as they provide new mechanistic insight into how the cytosolic bacterial pathogen Shigella flexneri evades IFN-induced host immunity.

*For correspondence:
jorn.coers@duke.edu

**Abstract** A central signal that marshals host defense against many infections is the lymphocyte-derived cytokine interferon-gamma (IFNγ). The IFNγ receptor is expressed on most human cells, and its activation leads to the expression of antimicrobial proteins that execute diverse cell-autonomous immune programs. One such immune program consists of the sequential detection, ubiquitylation, and destruction of intracellular pathogens. Recently, the IFNγ-inducible ubiquitin E3 ligase RNF213 was identified as a pivotal mediator of such a defense axis. RNF213 provides host protection against viral, bacterial, and protozoan pathogens. To establish infections, potentially susceptible intracellular pathogens must have evolved mechanisms that subdue RNF213-controlled cell-autonomous immunity. In support of this hypothesis, we demonstrate here that a causative agent of bacillary dysentery, *Shigella flexneri*, uses the type III secretion system (T3SS) effector IpaH1.4 to induce the degradation of RNF213. *S. flexneri* mutants lacking IpaH1.4 expression are bound and ubiquitylated by RNF213 in the cytosol of IFNγ-primed host cells. Linear (M1-) and lysine-linked ubiquitylation of *S. flexneri* requires RNF213 but is independent of the linear ubiquitin chain assembly complex (LUBAC). We find that ubiquitylation of *S. flexneri* is insufficient to kill intracellular bacteria, suggesting that *S. flexneri* employs additional virulence factors to escape from host defenses that operate downstream from RNF213-driven ubiquitylation. In brief, this study identified the bacterial IpaH1.4 protein as an inhibitor of mammalian RNF213 and highlights evasion of RNF213-driven immunity as a characteristic of the human-tropic pathogen *Shigella*.

## Introduction

The lymphocyte-derived cytokine interferon-gamma (IFNγ) is a strong inducer of cell-autonomous immunity (*Casanova et al., 2024*). The IFNγ-inducible E3 ubiquitin ligase ring finger protein 213 (RNF213) provides protection against a diverse group of intracellular pathogens that includes viruses, protozoa, and bacteria (*Hernandez et al., 2022*; *Otten et al., 2021*; *Walsh et al., 2022*; *Houzelstein et al., 2021*; *Thery et al., 2021*; *Matta et al., 2024*; *Martina et al., 2021*). RNF213 binds to bacteria that enter the host cell cytosol. Once bound to Gram-negative bacteria such as *Salmonella*, RNF213 directly ubiquitylates the bacterial surface molecule, lipopolysaccharide (LPS). In addition to RNF213, the linear ubiquitin chain assembly complex (LUBAC) was reported to play a critical role in the ubiquitylation of cytosolic bacteria (*Noad et al., 2017*). Whereas LUBAC catalyzes linear ubiquitin chains, the types of ubiquitin linkages deposited on bacterial surfaces by RNF213 are unknown. Following ubiquitylation, cytosolic bacteria can be captured by ubiquitin-binding proteins and delivered into autolysosomes for degradation (*Otten et al., 2021*).

While the predominantly vacuolar pathogen *Salmonella* is susceptible to RNF213-mediated cytosolic host defense (*Otten et al., 2021*), the cytosol-adapted Gram-negative pathogen *Burkholderia thailandensis* is resistant. *B. thailandensis* escapes RNF213-driven host defense in part through the secretion of the virulence factor TssM, a ubiquitin esterase that removes ubiquitin from LPS on the bacterial surface (*Szczesna et al., 2024*). However, whether and how other cytosolic bacterial pathogens escape from RNF213-driven ubiquitylation is unknown.

*Shigella* spp. are the etiological agent of bacillary dysentery and a leading cause of diarrheal deaths, malnutrition, and growth retardation in children (*Troeger et al., 2018*; *Libby et al., 2023*; *Bagamian et al., 2023*). *Shigella* spp. invade the intestinal epithelium, enter the host cell cytosol of infected human colonic epithelial cells, and trigger an acute inflammatory response that involves the secretion of IFNγ by lymphocytes (*Mellouk and Enninga, 2016*; *Levine et al., 2007*; *Raqib et al., 1995a*; *Raqib et al., 1995b*). IFNγ signaling in epithelial cells induces the expression of human guanylate binding protein 1 (GBP1), a defense protein that blocks bacterial intracellular motility and cell-to-cell spread (*Kutsch et al., 2021*; *Kutsch et al., 2020*; *Piro et al., 2017*; *Wandel et al., 2017*). To counter this potent cell-autonomous immune program, *Shigella flexneri* employs its type III secretion system (T3SS) to deliver the GBP1 inhibitor IpaH9.8 into the host cell cytosol. Loss of IpaH9.8 results in diminished *S. flexneri* virulence and a potent antibacterial host response (*Piro et al., 2017*; *Wandel et al., 2017*; *Goers et al., 2023*; *Li et al., 2017*).

IpaH9.8 belongs to a family of T3SS effectors that contain a leucine-rich repeat (LRR) domain involved in substrate recognition and a novel E3 ligase (NEL) domain that transfers K48-linked ubiquitin to its bound substrate (*Bullones-Bolaños et al., 2022*). K48-linked ubiquitylation tags the substrate for proteasomal degradation. These IpaH proteins belong to a 'second wave' of T3SS effectors that are under the control of the transcriptional regulator MxiE. Second wave T3SS effectors have been shown to promote host cell survival, to dampen inflammation, and to counteract cell-autonomous immune programs (*Schnupf and Sansonetti, 2019*), yet their full repertoire of targets remains to be determined.

In the present work, we found that the MxiE-regulated T3SS effector IpaH1.4 facilitates the proteasomal degradation of RNF213. Remarkably, RNF213 functions independent of LUBAC to promote the linear and lysine-linked ubiquitylation of *ipaH1.4*-deficient *S. flexneri* mutants. We observed that ubiquitylated *S. flexneri* continued to replicate inside the host cell cytosol, indicating that *S. flexneri* employs additional unknown virulence mechanisms to interfere with its degradation. Thus, our study demonstrates that the human enteric pathogen *S. flexneri* directly interferes with RNF213 expression and also limits host defense downstream of RNF213-driven ubiquitylation.

## Results

### Deletion of *mxiE* renders *S. flexneri* susceptible to linear ubiquitylation

Previous studies identified a cell-autonomous host defense program in which mammalian cells attach linear ubiquitin (M1) to the surface of the Gram-negative pathogen *Salmonella enterica* upon bacterial spillage into the host cell cytosol (*Otten et al., 2021*; *Noad et al., 2017*). Recently, it was reported that the cytosol-adapted pathogen *B. thailandensis* secretes a ubiquitin esterase that renders the bacteria resistant to this type of cytosolic ubiquitylation (*Szczesna et al., 2024*). We therefore hypothesized

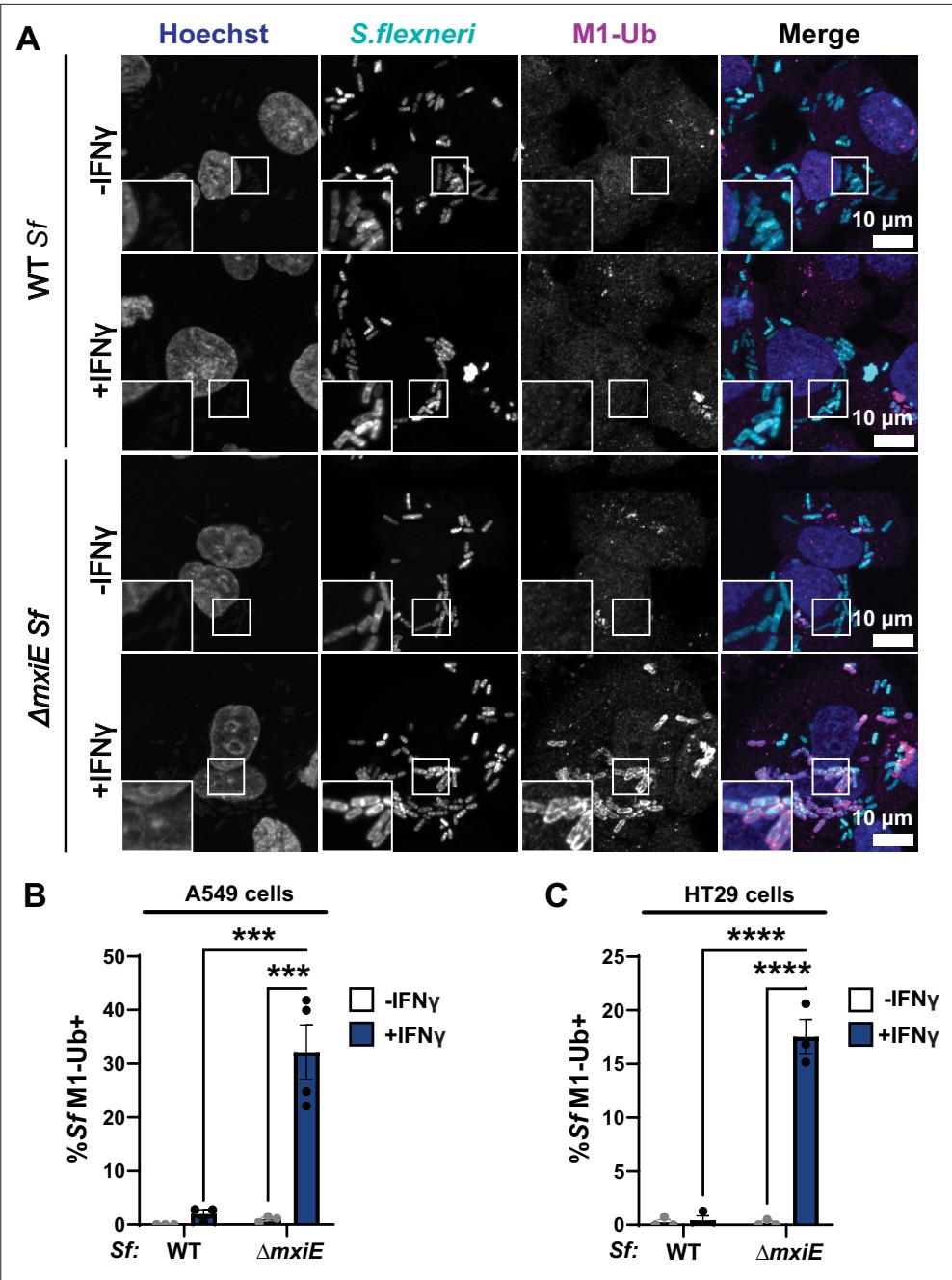

**Figure 1.** Mutation of the bacterial transcription factor MxiE results in ubiquitylation of *S. flexneri* in IFNγ-primed epithelial cells. (**A–C**) Cells were primed with 100 U/ml IFNγ or left unprimed and then infected with WT or *ΔmxiE S. flexneri* at an MOI of 50–100. Cells were fixed 3–4 hr post-infection (hpi) and stained for linear ubiquitin (M1-Ub) with anti-M1 antibody. Percentage of M1-Ub-positive bacteria was quantified in infected A549 epithelial cells at 4 hpi (**B**) and HT29 epithelial cells at 3 hpi (**C**). Graphs show the average of three independent experiments and depict means ± SEM. Two-way ANOVA with Tukey's multiple comparison tests was performed; all statistically significant comparisons are shown. ***p < 0.001; ****p < 0.0001; ns, not significant.

that resistance to host-driven linear ubiquitylation is a common feature of bacterial pathogens that are adapted to a cytosolic milieu. To test this concept, we monitored ubiquitylation of the professional cytosolic pathogen *S. flexneri* (serotype 2a, strain 257T) by immunofluorescence. We found that wild-type (WT) *S. flexneri* remained largely devoid of M1-linked ubiquitin in both alveolar A549 and colonic HT29 epithelial cell lines under both unprimed and IFNγ-primed conditions (*Figure 1A–C*).

In contrast to WT *S. flexneri*, a *ΔmxiE* mutant strain that has diminished expression of 'second-wave' T3SS effector proteins (*Kane et al., 2002*; *Mavris et al., 2002*), was robustly stained with anti-M1 antibodies in IFNγ-primed host cells (*Figure 1A–C*). These observations indicate that *S. flexneri* uses MxiE-dependent effectors to evade an IFNγ-activated ubiquitylation pathway that targets cytosolic bacteria.

## Linear and lysine-linked ubiquitin is attached to *S. flexneri ΔmxiE*

The ubiquitin protein contains seven lysine residues. Any of these lysine residues or the N-terminal methionine of the acceptor ubiquitin can form isopeptide bonds with the C-terminal glycine residue of the donor ubiquitin to form chains. To determine which ubiquitin linkage types are present on the surface of *S. flexneriΔmxiE* bacteria, we generated A549 cells that overexpressed internally tagged ubiquitin (INT-Ub) variants, which contain only a single lysine residue or lack lysines altogether. As a ubiquitin acceptor, the lysine-less (7KR) variant can only be assembled into ubiquitin chains via its N-terminal methionine. We found that 7KR INT-Ub robustly co-localized with *S. flexneri ΔmxiE*, albeit with diminished efficiency compared to WT INT-Ub (*Figure 2A*).

In addition to its incorporation into linear ubiquitin chains, 7KR ubiquitin can also function as donor ubiquitin and be attached as mono-ubiquitin to a substrate or to an existing ubiquitin chain as a chain terminator. To distinguish between 7KR INT-Ub signals originating from linear versus mono-ubiquitylation, we generated an N-terminally HA-tagged 7KR INT-Ub variant. The N-terminal tag prevents linear ubiquitylation but still allows 7KR INT-Ub to be attached as mono-ubiquitin via its C-terminal carboxyl group. We found that the addition of this N-terminal HA-tag significantly reduced but not completely abolished the number of *ΔmxiE* bacteria decorated with 7KR INT-Ub (*Figure 2 – figure supplement 1*). These data indicate that much of the association of 7KR INT-Ub with cytosolic *ΔmxiE* bacteria is due to the formation of M1-linked linear ubiquitin chains.

Collectively, our data confirmed that M1-linked ubiquitin is bound to *S. flexneri ΔmxiE* but also indicated that other additional ubiquitin linkage types are attached to *mxiE*-deficient bacteria. The functional importance of these lysine-linked ubiquitin chains was demonstrated by improved bacterial targeting of INT-Ub variants that can form additional K27-linkages or both K27- and K63-linkages (*Figure 2A*). Linkage-specific antibodies against K27 and K63 confirmed that these two linkage types are present on *ΔmxiE* bacteria in IFNγ-primed A549 cells (*Figure 2B, C*). Staining with an antibody that detects most ubiquitin linkage types (FK2) showed near complete overlap with anti-M1 staining (*Figure 2D, E*). Because the percentage of FK2-positive but M1-negative bacteria (*Figure 2E*) was much smaller than the percentage of bacteria that were K27- or K63-positive (*Figure 2B, C*), these data suggest that most *S. flexneri ΔmxiE* bacteria are decorated with both linear and lysine-linked ubiquitin.

## The ubiquitylation of *S. flexneri ΔmxiE* is dependent on RNF213 but not LUBAC

Previous work showed that the multimeric E3 ligase LUBAC can deposit M1-linked ubiquitin on the surface of cytosolic *S. enterica* Typhimurium (*Noad et al., 2017*). Moreover, *S. flexneri* secretes the MxiE-regulated effector IpaH1.4 which specifically recognizes and marks the LUBAC components HOIP and HOIL-1 for proteasomal degradation (*de Jong et al., 2016*; *Liu et al., 2022*; *Hiragi et al., 2023*). Consequently, we hypothesized that LUBAC was essential for the ubiquitylation of *ΔmxiE*. To test this hypothesis, we infected HOIP- and HOIL-1-deficient A549 cells (HOIP^KO, HOIL-1^KO, *Figure 3A*) with the *ΔmxiE* mutant under IFNγ-primed conditions. Unexpectedly, deletion of either HOIP or HOIL-1, each an essential component of the LUBAC complex, did not diminish linear ubiquitylation of *ΔmxiE* when compared to WT cells (*Figure 3B*). Similarly, the internally tagged 7KR-Ub variant decorated *S. flexneri ΔmxiE* independent of the LUBAC component HOIP (*Figure 3C*). Collectively, these data demonstrate that LUBAC is dispensable for linear ubiquitylation of *ΔmxiE* bacteria in the host cell cytosol of IFNγ-primed A549 cells.

The ubiquitin E3 ligase RNF213 was recently shown to ubiquitylate pathogen-containing vacuoles (*Hernandez et al., 2022*; *Walsh et al., 2022*) as well as Gram-positive and Gram-negative bacteria inside the host cell cytosol (*Otten et al., 2021*; *Thery et al., 2021*; *Szczesna et al., 2024*). Like HOIL-1, RNF213 is an IFNγ-inducible protein (*Figure 3A* and *Hernandez et al., 2022*; *Walsh et al., 2022*). Using immunofluorescence microscopy, we detected RNF213 on the surface of *ΔmxiE* but

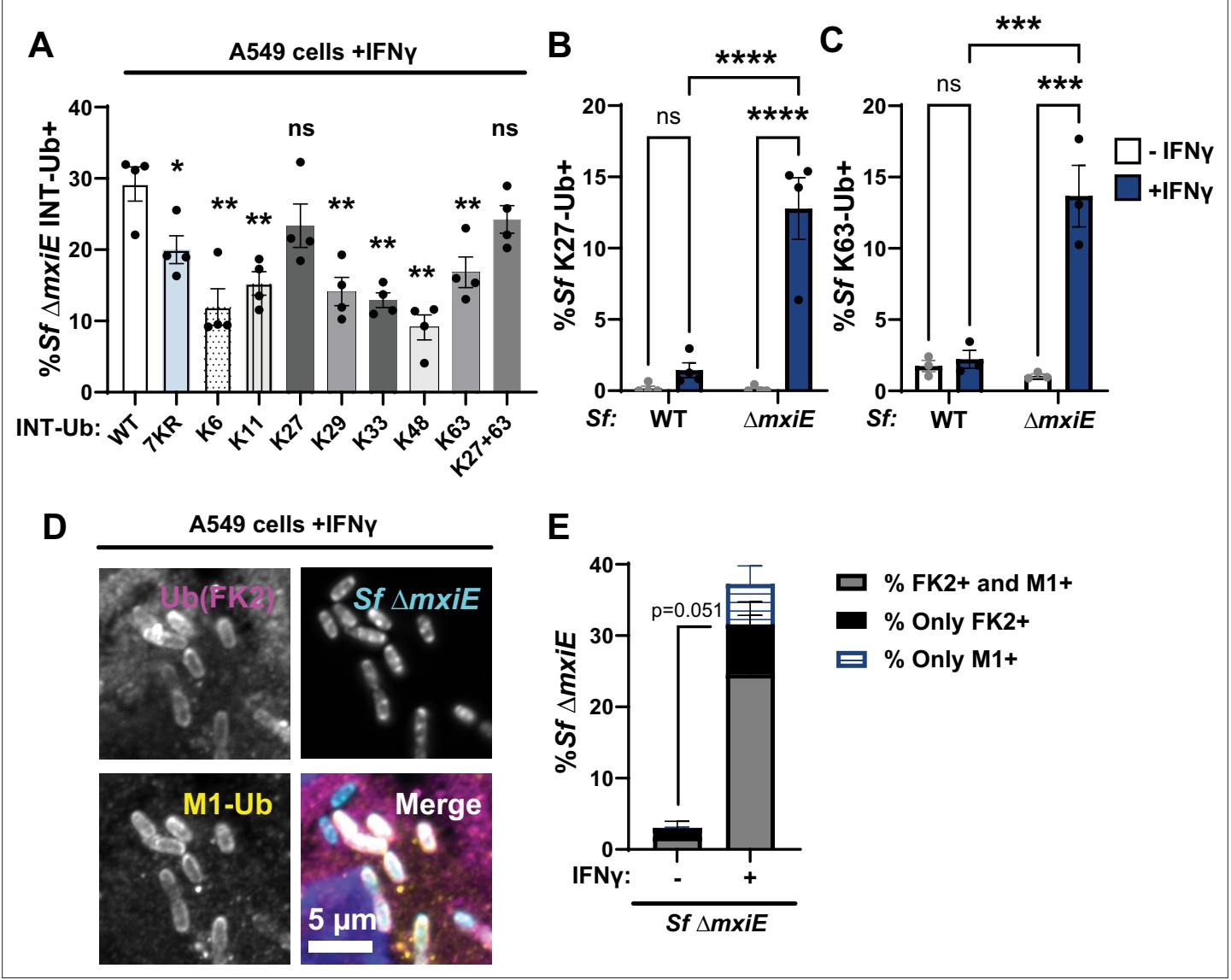

**Figure 2.** *S. flexneri ΔmxiE* mutants become decorated with linear and lysine-linked ubiquitin in IFNγ-primed A549 cells. (**A**) WT A549 cells expressing the indicated internally Strep-tagged ubiquitin (INT-Ub) variants were primed with 100 U/ml IFNγ overnight and co-localization of INT-Ub with cytosolic *S. flexneri* (Sf) *ΔmxiE* was assessed at 4 hpi. Ubiquitin linkage-specific antibodies were used to determine the percentage of *S. flexneri ΔmxiE* staining positive for K27-linked ubiquitin (K27-Ub) (**B**) or K63-linked ubiquitin (**C**) in IFNγ-primed untransduced WT A549 cells. Co-staining of anti-M1 and anti-ubiquitin (FK2) on the surface of *S. flexneri ΔmxiE* in IFNγ-primed (100 U/ml) A549 cells is shown in (**D**). Quantification of anti-M1-FK2 co-staining in untreated and IFNγ-primed A549 cells is depicted in (**E**). All infections were done at an MOI of 50–100. Data were generated from at least three independent experiments and show mean ± SEM. One-way ANOVA followed by Dunnett's multiple comparisons was performed of all groups against WT-ubiquitin group (**A**). Two-way ANOVA with Tukey's multiple comparison tests was performed for (**B**) and (**C**). An unpaired *t*-test was performed between 'both FK2 + M1' groups (gray bars) (**E**). *p < 0.05; **p < 0.01; ***p < 0.001; ****p < 0.0001.

The online version of this article includes the following figure supplement(s) for figure 2:

**Figure supplement 1.** Addition of an N-terminal HA-tag significantly reduces targeting of 7KR ubiquitin to *S. flexneri* ΔmxiE.

not WT bacteria in IFNγ-primed A549 as well as colonic HT29 cells (***Figure 3D–G***). The vast majority of ubiquitylated *S. flexneri ΔmxiE* mutants (~93%) co-stained with anti-RNF213 (***Figure 3G***), indicative of a functional role for RNF213 in the ubiquitylation of *ΔmxiE* bacteria. To directly probe for RNF213 function, we infected a pool of CRISPR-generated A549 RNF213[KO] cells (***Figure 3A***) with *S. flexneri ΔmxiE* and stained these IFNγ-primed cells for M1-, K27-, and K63-linked ubiquitin. We found that in the absence of human RNF213, antibodies specific for M1-, K27-, and K63-linked ubiquitin no longer stained the surface of *S. flexneri ΔmxiE* (***Figure 3B, H–I***). In agreement with these

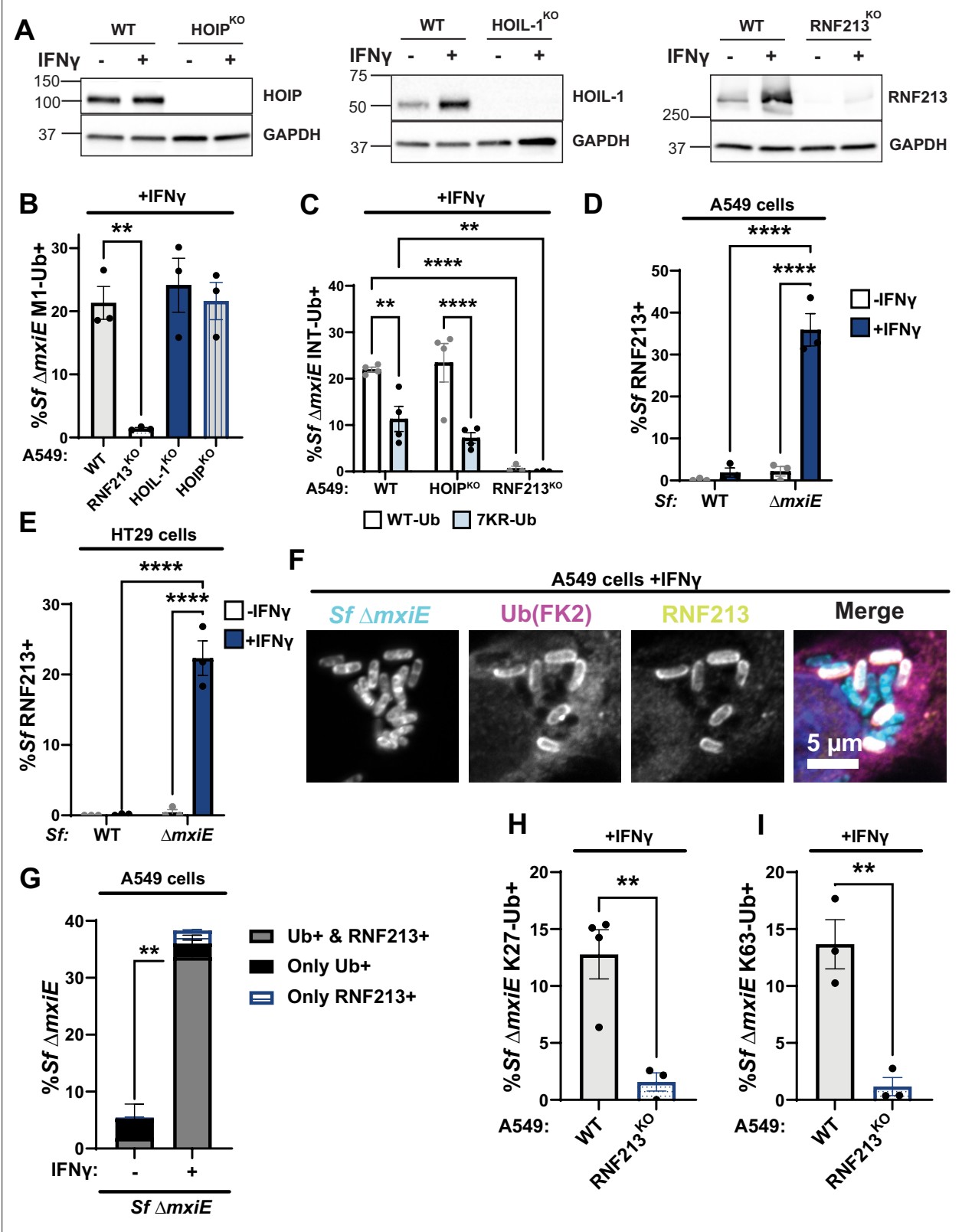

**Figure 3.** Ubiquitylation of *S. flexneri ΔmxiE* is dependent on RNF213 but not LUBAC. (**A**) Immunoblotting for HOIP, HOIL-1, and RNF213 protein expression in untreated and IFNγ-primed WT and the corresponding gene deletion (KO) A549 cells. (**B**) Percentage *ΔmxiE S. flexneri* bacteria stained with anti-M1-linked ubiquitin in IFNγ-primed WT, HOIP^KO, HOIL-1^KO, and RNF213^KO A549 cells. (**C**) Percentage of WT and 7KR INT-Ub-positive *ΔmxiE S. flexneri* in IFNγ-primed WT, HOIP^KO, and RNF213^KO A549 cells. (**D–G**) Untreated and IFNγ-primed A549 and HT29 cells were infected with the

*Figure 3 continued on next page*

Figure 3 continued

indicated *S. flexneri* strains and immuno-stained for RNF213 and ubiquitin (FK2). Representative immunofluorescence microscopy images are shown for A549 infections in (**F**). RNF213-*S. flexneri* colocalization percentages were quantified in A549 (**D**) and HT29 (**E**) cells. (**G**) Quantification of ubiquitin and RNF213 colocalization with Δ*mxiE S. flexneri* in A549 cells. Percentages of Δ*mxiE S. flexneri* staining positive for antibodies specific for K27-linked (**H**) and K63-linked ubiquitin (**I**) are also provided. All data are represented by the mean ± SEM from at least three independent experiments. One-way ANOVA followed by Dunnett's multiple comparisons was performed of all groups against WT A549 group in (**A**). Two-way ANOVA with Tukey's multiple comparison tests was performed in (**C, D, F**). An unpaired *t*-test was performed between the 'Ub+ and RNF213+' groups (gray bars) in (**G**). For (**H, I**), an unpaired *t*-test was performed. Comparisons not shown are non-significant. **$p < 0.01$; ****$p < 0.0001$.

The online version of this article includes the following source data for figure 3:

**Source data 1.** PDF file containing original western blots for **Figure 3A**, indicating the relevant bands and treatments.

**Source data 2.** Original files for western blot analysis displayed in **Figure 3A**.

immunofluorescence data, both WT and lysine-less ubiquitin (7KR-Ub) localized to Δ*mxiE* bacteria in WT but not in RNF213$^{KO}$ cells (**Figure 3C**). Together, these data demonstrate that human RNF213 is essential for the linear and lysine-linked ubiquitylation of MxiE-deficient *S. flexneri* mutants.

## Ectopic expression of *S. flexneri* virulence factors IpaH1.4 or IpaH2.5 induces RNF213 degradation

The transcription factor MxiE induces the expression of several secreted effector proteins of the IpaH family that act as bacterial ubiquitin E3 ligases and tag host proteins for proteasomal degradation (**Wandel et al., 2017**; **Li et al., 2017**; **de Jong et al., 2016**; **Liu et al., 2022**; **Luchetti et al., 2021**; **Hansen et al., 2021**; **Otsubo et al., 2019**; **Rohde et al., 2007**; **Ashida et al., 2010**). Since our data revealed that the *S. flexneri* virulence factor MxiE blocks bacterial ubiquitylation by RNF213, we hypothesized that *S. flexneri* infections induce the proteasomal degradation of RNF213 in a MxiE-dependent manner. To test this hypothesis, we measured RNF213 protein levels by immunoblotting in cells that had been infected with WT or Δ*mxiE S. flexneri*. Our data revealed that RNF213 protein levels are drastically reduced upon infection with WT *S. flexneri* and that the deletion of *mxiE* partially restored RNF213 expression in infected cells (**Figure 4A**). Similarly, and as reported previously (**Noad et al., 2017**; **de Jong et al., 2016**; **Liu et al., 2022**), HOIP protein expression is diminished in cells infected with WT *S. flexneri* but much less so in the case of infections with Δ*mxiE S. flexneri* (**Figure 4A**).

To directly test whether *S. flexneri* promotes proteasomal degradation of RNF213, we conducted infections in the presence of the proteasomal inhibitor MG132. As expected, MG132 largely restored RNF213 total protein levels in *S. flexneri*-infected cells (**Figure 4B**). We next tested whether a specific IpaH effector facilitates the degradation of RNF213. To this end, we transiently expressed GFP-fusion proteins of individual IpaH family members in HEK 293T cells that ectopically expressed RNF213 and measured their effects on RNF213 protein levels by immunoblotting. Our mini-screen revealed that the ectopic expression of the two closely related paralogs IpaH1.4 and IpaH2.5 was sufficient to promote RNF213 degradation in HEK 293T cells (**Figure 4C**). Of note, the substrate-binding LRR domains of IpaH2.5 and IpaH1.4 are 98.9% identical in DNA sequence and were previously shown to have comparable substrate specificity and functional redundancy (**de Jong et al., 2016**). Both IpaH1.4 and 2.5 share a highly conserved NEL E3 ubiquitin ligase domain that contains an essential catalytic cysteine residue at position 368 (**Liu et al., 2022**). Catalytically inactive forms of IpaH1.4 or IpaH2.5, in which Cys368 is mutated to a serine (C/S), failed to degrade RNF213 (**Figure 4D**). As an additional control, we included overexpression of IpaH9.8, an inhibitor of GBPs, which also did not impact RNF213 expression (**Figure 4D**), as anticipated.

To investigate potential physical interactions between IpaH1.4 and IpaH2.5, we reanalyzed a previously generated dataset that employed a method known as ubiquitin-activated interaction traps (UBAITs) (**Hansen et al., 2021**). As depicted in **Figure 4—figure supplement 1A**, the human ubiquitin gene was fused to the 3' end of IpaH2.5, producing a C-terminal IpaH2.5-ubiquitin fusion protein. When incubated with ATP, a ubiquitin-activating enzyme E1, and a ubiquitin-conjugating enzyme E2, the IpaH2.5-ubiquitin 'bait' protein is capable of binding to and ubiquitylating target substrates. This ubiquitylation creates an iso-peptide bond between the IpaH2.5 bait and its substrate, thereby enabling purification via a Strep affinity tag incorporated into the fusion construct (**Hansen et al., 2021**). IpaH2.5-ubiquitin bait and IpaH3-ubiquitin control proteins were incubated with lysates from murine intestinal tissue. To detect interaction partners in a physiologically relevant setting, we used

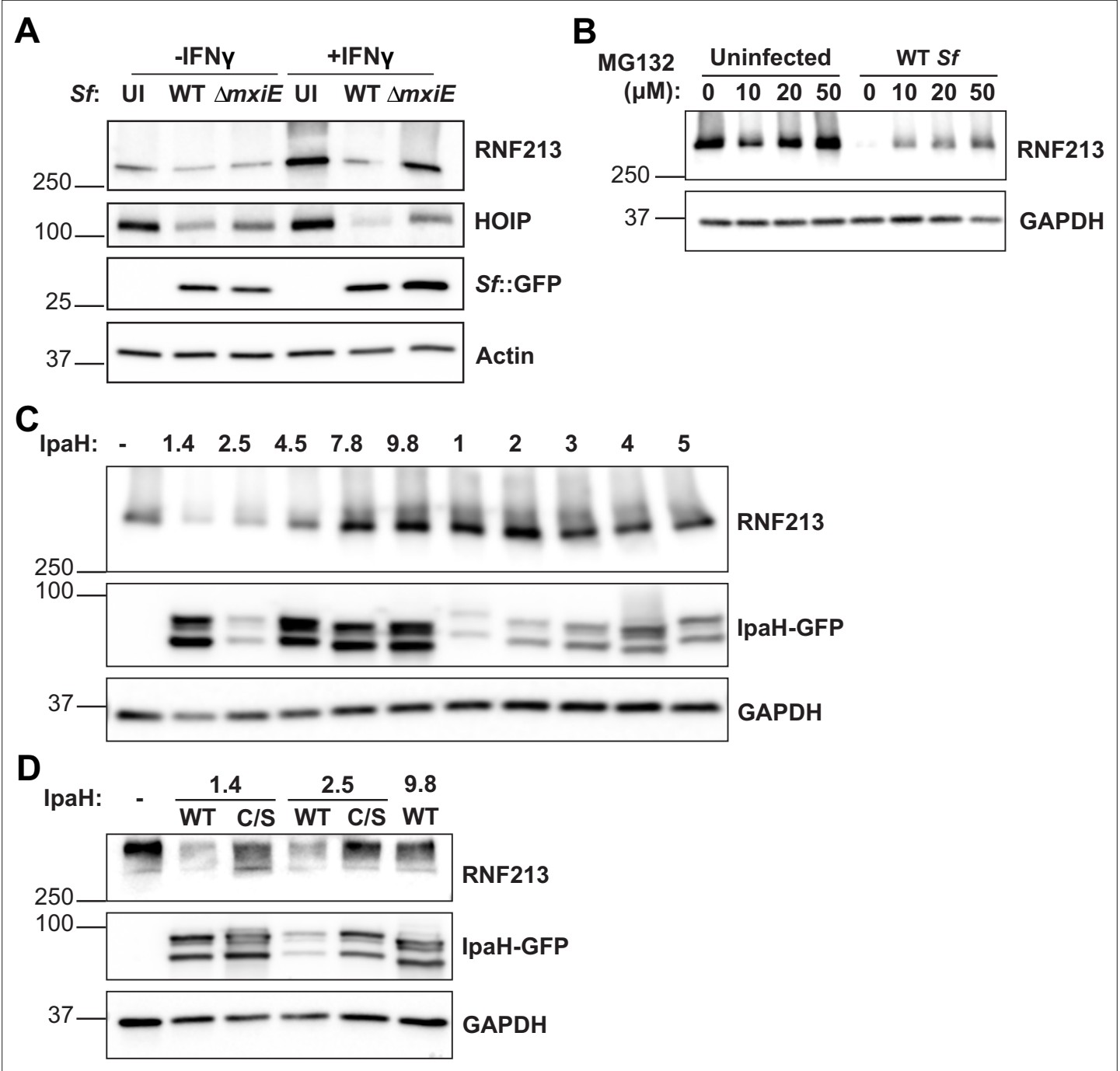

**Figure 4.** *S. flexneri* virulence factors IpaH1.4 and IpaH2.5 induce proteasomal degradation of RNF213 (**A, B**). All *S. flexneri* strains express PilT to enhance adhesion and infection rates, and infections were carried out at an MOI of 5–25 for 3 hr. (**A**) Untreated and IFNγ-primed (100 U/ml) A549 cells were infected with indicated *S. flexneri* strains expressing GFP. Protein lysates were probed for expression of RNF213, HOIP, and GFP. GFP expression serves as a bacterial protein loading control. (**B**) Infected and uninfected A549 cells were cultured in the presence of different concentrations of the proteasomal inhibitor MG132 and RNF213 expression was monitored by immunoblotting. (**C**) HEK 293T cells stably expressing mCherry-RNF213 were transiently transfected with individual GFP-tagged IpaH effectors for 24 hr and RNF213 expression was assessed. (**D**) WT and catalytically inactive C368S mutants of IpaH1.4 and IpaH2.5 were transiently transfected into HEK 293T cells expressing mCherry-RNF213 and cell lysates were subjected to immunoblotting. (**C, D**) Denaturation of cell lysates at lower temperature (56°C) was required for RNF213 detection but also resulted in the formation of double bands for all IpaH-GFP constructs. Images are representative of three independent experiments. UI: uninfected.

The online version of this article includes the following source data and figure supplement(s) for figure 4:

**Source data 1.** PDF file containing original western blots for *Figure 4*, indicating the relevant bands and treatments.

*Figure 4 continued on next page*

Figure 4 continued

**Source data 2.** Original files for western blot analysis displayed in *Figure 4*.

**Figure supplement 1.** Ubiquitin-activated interaction trap (UBAIT) identifies mouse Rnf213 as likely interaction partner of IpaH2.5.

intestinal lysates derived from mice infected with *Salmonella,* which in contrast to *Shigella* causes pronounced inflammation in WT mice and therefore better simulates human *Shigellosis* in an animal model. Using UBAIT, we identified HOIP (Rnf31) as a likely IpaH2.5 binding partner (*Figure 4—figure supplement 1B*), thus confirming previous observations (*de Jong et al., 2016*) and validating the effectiveness of our approach. Strikingly, we identified mouse Rnf213 as the most abundant interaction partner of the IpaH2.5-ubiquitin bait protein (*Figure 4—figure supplement 1B*). Collectively, our data and concurrent reports showing direct interactions between IpaH1.4 and human RNF213 (*Naydenova et al., 2025*; *Zhou et al., 2025*) indicate that the virulence factors IpaH1.4 and IpaH2.5 directly bind and degrade mouse as well as human RNF213.

## Loss of IpaH1.4 is sufficient to render *S. flexneri* susceptible to RNF213-driven ubiquitylation in human and murine host cells

Given that ectopic expression of either IpaH1.4 or IpaH2.5 was sufficient to degrade RNF213, we expected these two endogenous effectors to be functionally redundant. However, we found that genetic deletion of *ipaH1.4* in either *S. flexneri* serotype 2a strain 257T (*Figure 5A, B*) or in serotype 5a strain M90T (*Figure 5—figure supplement 1A*) was sufficient to restore RNF213 protein expression and RNF213 recruitment to cytosolic bacteria. A deletion mutant of *ipaH2.5* was as resistant to RNF213 targeting as co-isogenic WT bacteria, and a Δ*ipaH1.4-ipaH2.5* double mutant was indistinguishable from the Δ*ipaH1.4* single deletion mutant (*Figure 5—figure supplement 1A*). These data suggested that endogenous IpaH2.5 is non-functional, which agrees with a previous characterization of *ipaH2.5* as a minimally expressed pseudogene (*Silué et al., 2020*).

Previous work showed that IpaH1.4 binds to and degrades the LUBAC subunit HOIP (*Noad et al., 2017*; *de Jong et al., 2016*; *Liu et al., 2022*). To determine whether the degradation of RNF213 could be indirect and HOIP-dependent, or vice versa, we monitored RNF213 expression in HOIP[KO] cells and HOIP expression in RNF213[KO] cells. We found that expression of RNF213 remained at WT levels in the absence of HOIP and that the degree of IpaH1.4-dependent RNF213 degradation was comparable in WT and HOIP[KO] cells (*Figure 5A*). Similarly, infections with WT but not Δ*ipaH1.4* mutant *S. flexneri* resulted in a substantial reduction of HOIP protein expression in RNF213[KO] cells (*Figure 5—figure supplement 1B*). These observations show that IpaH1.4 targets RNF213 and HOIP for degradation independent from each other.

The sustained RNF213 expression in cells infected with *S. flexneri*Δ*ipaH1.4* (*Figure 5A*) correlated with staining of cytosolic Δ*ipaH1.4* bacteria with RNF213 (*Figure 5B, C*). As expected, the complementation of the Δ*ipaH1.4* mutant with WT IpaH1.4 efficiently prevented RNF213 recruitment to cytosolic bacteria, whereas complementation with catalytically inactive IpaH1.4 did not (*Figure 5C, D*). Through the use of linkage-specific antibodies, we also observed M1-, K27-, and K63-linked ubiquitin on cytosolic Δ*ipaH1.4* bacteria in IFNγ-primed WT A549 cells (*Figure 5E*; *Figure 5—figure supplement 1C–E*). In contrast to IFNγ-primed WT A549 cells, IFNγ-primed RNF213[KO] A549 cells lacked linear ubiquitylation of Δ*ipaH1.4* (*Figure 5E*). This ubiquitylation of *S. flexneri*Δ*ipaH1.4* was independent of LUBAC, as it occurred in HOIL-1[KO], HOIP[KO], and WT A549 cells at comparable frequencies (*Figure 5—figure supplement 1F*).

Although *S. flexneri* is a human-adapted pathogen, some IFNγ-inducible host defense programs targeting *S. flexneri* and the corresponding bacterial evasion mechanisms transcend host barriers. For example, the *Shigella* virulence factor IpaH9.8 inhibits both human and murine guanylate binding proteins (GBPs) (*Li et al., 2017*; *Ji et al., 2019*). We therefore asked whether RNF213 ubiquitylates *S. flexneri* Δ*ipaH1.4* in at least one other mammalian species, the mouse. To do so, we infected unprimed and IFNγ-primed WT and Rnf213[KO/KO] mouse embryonic fibroblasts (MEFs) (*Figure 5—figure supplement 2A*) with *S. flexneri* WT and Δ*ipaH1.4* mutant. We observed that Δ*ipaH1.4* bacteria became decorated with mouse Rnf213 (*Figure 5—figure supplement 2B*) and linear ubiquitin chains (*Figure 5—figure supplement 2C*). The IFNγ-dependent increase in the percentage of M1-ubiquitin-decorated Δ*ipaH1.4* *S. flexneri* bacteria was reversed in *Rnf213*[KO/KO] MEFs (*Figure 5—figure*

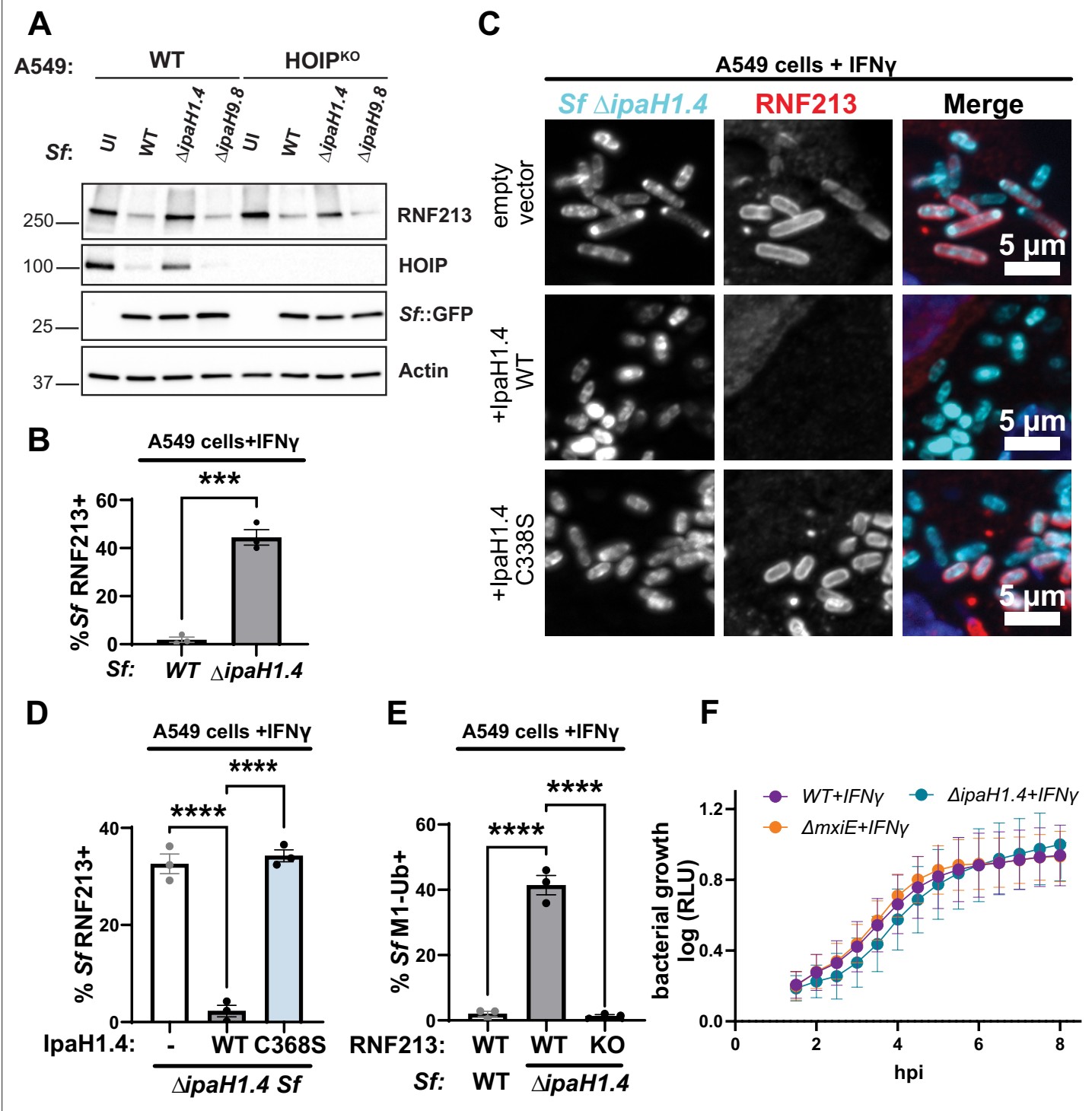

**Figure 5.** Loss of IpaH1.4 is sufficient to render *S. flexneri* susceptible to RNF213-driven ubiquitylation. (**A**) IFNγ-primed WT and HOIP^KO A549 cells were infected with the indicated PilT⁺, GFP⁺*S. flexneri* strains for 3 hr at an MOI of 25 and RNF213 protein levels were assessed by Western blotting. Bacterial GFP and host actin are used as host protein loading controls. (**B**) Percentages of RNF213⁺ WT and Δ*ipaH1.4 S. flexneri* in IFNγ-primed A549 cells are shown. (**C**) Representative microscopy images depicting RNF213 recruitment to cytosolic Δ*IpaH1.4* bacteria complemented with empty vector, WT IpaH1.4, or catalytically inactive IpaH1.4 (C368S) in IFNγ-primed WT A549 cells. (**D**) Percentages of RNF213⁺ indicated *S. flexneri* strains in IFNγ-primed WT A549 cells are shown. (**E**) Percentages WT and Δ*IpaH1.4 S. flexneri* stained positive for anti-M1-linked antibody in IFNγ-primed WT and RNF213^KO A549 cells are depicted. (**F**) The ilux operon was introduced into *S. flexneri* strains to use bioluminescence relative light units (RLU) as a proxy for bacterial growth. RLU was measured in IFNγ-primed WT A549 cells infected with the indicated *S. flexneri* strains at an MOI of 5. (**B, D, E**) Data represent

*Figure 5 continued on next page*

*Figure 5 continued*

the mean ± SEM from at least three independent experiments. An unpaired *t*-test was conducted for (**B**). One-way ANOVA followed by Dunnett's multiple comparisons was performed between all groups against the group Δ*ipaH1.4* expressing WT IpaH1.4 (**D**) and WT A549 cells (**E**). ***p < 0.001; ****p < 0.0001.

The online version of this article includes the following source data and figure supplement(s) for figure 5:

**Source data 1.** PDF file containing original western blots for *Figure 5A*, indicating the relevant bands and treatments.

**Source data 2.** Original files for western blot analysis displayed in *Figure 5A*.

**Figure supplement 1.** Deletion of *ipaH1.4* but not *ipaH2.5* results in RNF213 recruitment to cytosolic *S. flexneri*.

**Figure supplement 1—source data 1.** PDF file containing original western blots for *Figure 5—figure supplement 1B*, indicating the relevant bands and treatments.

**Figure supplement 1—source data 2.** Original files for western blot analysis displayed in *Figure 5—figure supplement 1B*.

**Figure supplement 2.** Mouse Rnf213 binds to Δ*ipaH1.4 S. flexneri* and facilitates linear ubiquitylation of cytosolic bacteria.

**Figure supplement 2—source data 1.** PDF file containing original western blots for *Figure 5—figure supplement 2A*, indicating the relevant bands and treatments.

**Figure supplement 2—source data 2.** Original files for western blot analysis displayed in *Figure 5—figure supplement 2A*.

supplement 1C). Collectively, these data depict a conserved role for the mammalian ubiquitin E3 ligase RNF213 in cytosolic immune surveillance and demonstrate that IpaH1.4 can evade both mouse and human RNF213-driven linear ubiquitylation.

Ubiquitylated microbial cargo can be delivered into microbicidal lysosomes in an autophagy-like process commonly referred to as xenophagy (*Tripathi-Giesgen et al., 2021*). Previous work showed that linear ubiquitylation of *S. enterica* leads to bacterial clearance by xenophagy (*Otten et al., 2021*). In contrast to the xenophagic killing of ubiquitylated *S. enterica*, we found that ubiquitylated *S. flexneri ΔIpaH1.4* and *ΔmxiE* mutant bacteria continued to replicate inside IFNγ-primed A549 cells (*Figure 5H*), reminiscent of the continued growth of ubiquitylated *B. thailandensis tssM* mutants inside the host cell cytosol of human IFNγ-primed cells (*Szczesna et al., 2024*). Thus, our data indicate that *S. flexneri*, like *B. thailandensis*, can block ubiquitin-dependent xenophagy both at the stage of ubiquitylation and at another so far undetermined downstream step of this cell-autonomous host defense pathway.

## Discussion

Xenophagy is a cell-autonomous defense program in which intracellular pathogens are captured and delivered into degradative lysosomes. Ubiquitylation of intracellular pathogens is the most common initiating step for xenophagy (*Tripathi-Giesgen et al., 2021*). Like other steps of the xenophagy cascade, pathogen ubiquitylation is augmented in cells primed with the lymphocyte-derived cytokine IFNγ (*Hernandez et al., 2022*; *Walsh et al., 2022*; *Matta et al., 2024*; *Haldar et al., 2015*; *Fernandez et al., 2024*). An antimicrobial E3 ubiquitin ligase, RNF213, is itself an IFNγ-inducible protein (*Hernandez et al., 2022*; *Otten et al., 2021*; *Walsh et al., 2022*; *Houzelstein et al., 2021*; *Thery et al., 2021*; *Matta et al., 2024*). Here, we report that the enteric human pathogen *S. flexneri* employs the T3SS effector IpaH1.4 to induce the proteasomal degradation of RNF213. Accordingly, loss of IpaH1.4 results in RNF213-dependent ubiquitylation of *S. flexneri*. However, IpaH1.4-deficient *S. flexneri* mutants, despite their ubiquitylation, continue to replicate inside the cytosol of IFNγ-primed human cells. These observations suggest that *S. flexneri* not only evades RNF213-driven ubiquitylation but also the downstream degradation of ubiquitylated bacteria.

Accompanying previous reports on *Chlamydia* and *Burkholderia* (*Walsh et al., 2022*; *Szczesna et al., 2024*), our study on *Shigella* provides evidence that host-adapted intracellular pathogens escape from RNF213-driven host defense. Each of these pathogens evolved a mechanistically distinct strategy to escape ubiquitylation by RNF213. The soil-dwelling accidental pathogen *B. thailandensis* is recognized and ubiquitylated by RNF213 but instantly removes ubiquitin from its surface through the activity of its secreted ubiquitin esterase TssM (*Szczesna et al., 2024*). The human pathogen *Chlamydia trachomatis* decorates its surrounding vacuole with the secreted T3SS effector GarD, which renders the *Chlamydia*-containing vacuole invisible to RNF213 (*Walsh et al., 2022*). In contrast to TssM and GarD, the *Shigella* effector IpaH1.4 is a direct inhibitor of RNF213, which is further supported by

recently published biochemical studies demonstrating direct interactions between the LRR domain of IpaH1.4 and the RING domain of RNF213 (*Naydenova et al., 2025*; *Zhou et al., 2025*).

IpaH1.4 and IpaH2.5 have nearly 99% sequence identity in their substrate-binding LRR domains, and overexpression of either paralog is sufficient to trigger RNF213 degradation. However, only IpaH1.4 appears to be essential for *S. flexneri*-triggered RNF213 degradation – as shown in the present work – or for HOIP degradation (*de Jong et al., 2016*). While these observations agree with a previous report demonstrating low expression of IpaH2.5 under standard culture conditions (*Silué et al., 2020*), it remains to be seen whether IpaH2.5 is expressed and possibly redundant with IpaH1.4 for virulence during in vivo infections.

The mechanism by which IpaH1.4 and its paralog IpaH2.5 promote RNF213 degradation can be gleaned from previous studies characterizing IpaH1.4 and IpaH2.5 as ubiquitin E3 ligases that directly conjugate K48-linked ubiquitin to the LUBAC subunit HOIP leading to HOIP degradation by the proteasome (*de Jong et al., 2016*). Indeed, studies published during the revisions of our manuscript show that IpaH1.4 and IpaH2.5 similarly bind to and directly ubiquitylate RNF213 to facilitate RNF213's proteasomal degradation (*Naydenova et al., 2025*; *Zhou et al., 2025*). Such direct interactions are additionally supported by our data showing that transient IpaH2.5–RNF213 interactions can be trapped via formation of covalent bonds between these proteins in a UBAIT assay.

The deletion of *ipaH1.4* leads to *S. flexneri* ubiquitylation in IFNγ-primed host cells. Different ubiquitin linkage types and their incorporation into heterotypic conjugates create a rich diversity of ubiquitin modifications with unique functional outputs, which is commonly referred to as the 'ubiquitin code.' (*Komander and Rape, 2012*; *Dikic and Schulman, 2023*). We found that at least three linkage types are attached to *S. flexneri* Δ*ipaH1.4* in an RNF213-dependent manner: K27, K63, and linear (M1). Linear ubiquitylation is a head-to-tail inter-ubiquitin linkage through the amino group of the N-terminal methionine (M1) and the C-terminal carboxyl group (*Kirisako et al., 2006*). The LUBAC complex is the only recognized ubiquitin E3 ligase able to conjugate M1-linked ubiquitin chains (*Sasaki and Iwai, 2023*; *Gao et al., 2023*). However, we observed that LUBAC is dispensable for M1-linked ubiquitylation of cytosolic *S. flexneri*Δ*ipaH1.4*. We found that lysine-less internally tagged ubiquitin or an M1-specific antibody bound to *S. flexneri* Δ*ipaH1.4* in cells lacking LUBAC (HOIL-1[KO] or HOIP[KO]) but failed to bind bacteria in RNF213-deficient cells. These data corroborate our previous observations, which suggested that IFNγ-induced linear ubiquitylation of *Chlamydia*- and *Toxoplasma*-containing vacuoles occurs independent of LUBAC in human cells (*Hernandez et al., 2022*; *Walsh et al., 2022*). While RNF213 itself may catalyze such linear ubiquitin conjugation, it is equally possible that this reaction is carried out by additional ubiquitin E3 ligases that are being recruited to the bacterial surface alongside RNF213.

It was previously reported that RNF213 ubiquitylates LPS on the surface of *S. enterica*, prompting the recruitment of LUBAC, which catalyzes the conjugation of linear ubiquitin to the surface of *S. enterica* (*Otten et al., 2021*). Whereas LUBAC was found to be essential for M1-linked ubiquitylation of cytosolic *S. enterica* (*Noad et al., 2017*), we find that LUBAC is dispensable for M1-linked ubiquitylation of *S. flexneri*Δ*ipaH1.4* bacteria. While differences between *S. enterica* and *S. flexneri*, for example, in the composition of their bacterial cell walls, could possibly contribute to these discrepant results, it is also important to note that the previous *S. enterica* infection studies were conducted in naïve cells (*Otten et al., 2021*; *Noad et al., 2017*), whereas our studies were performed under IFNγ priming conditions. We therefore propose that a boost in the expression of RNF213 and other IFN-stimulated genes (ISGs) renders LUBAC obsolete for M1-linked ubiquitylation of Gram-negative bacteria. Future studies are needed to determine how RNF213 co-operates with LUBAC and other host factors to ubiquitin-tag intracellular pathogens.

*Shigella* is a human-adapted pathogen that secretes T3SS effectors to directly interfere with type I and III IFN signaling in epithelial cells (*Alphonse et al., 2022*). Additionally, *Shigella* employs virulence factors that inhibit ISGs which act downstream from IFN signaling. For example, the T3SS effector IpaH9.8 blocks the activity of IFNγ-inducible antimicrobial ISGs such as GBP1 (*Piro et al., 2017*; *Wandel et al., 2017*; *Goers et al., 2023*; *Li et al., 2017*). Here, we report that IpaH1.4 inactivates another ISG, that is RNF213. Because ubiquitylated *S. flexneri* Δ*ipaH1.4* and Δ*mxiE* mutants continue to replicate inside the cytosol of IFNγ-primed host cells, we propose that *S. flexneri* also uses a mxiE-independent T3SS effector to interfere with host defenses occurring downstream from RNF213. Collectively, these data indicate that *Shigella* is equipped with redundant counter-immune

mechanisms. This type of bacterial effector redundancy is not unique to *Shigella*. It was recently shown that the rodent pathogen *Citrobacter rodentium* can tolerate various combinations of individual T3SS effector deletions in vivo, an observation that led to the 'T3SS effector networks' hypothesis (*Ruano-Gallego et al., 2021*; *Sanchez-Garrido et al., 2022*). Further characterization of the *Shigella* T3SS effector network that renders bacteria resistant to IFNγ-inducible cell-autonomous immune program will provide a deeper understanding of the pathogenesis of this important human enteric pathogen. Given the rise of extensively drug-resistant *Shigella* strains (*Tsai et al., 2024*; *Donkor et al., 2024*; *Matanza and Clements, 2023*; *O'Flanagan et al., 2023*), such an understanding may prove to be essential for the development of critically needed novel therapeutic approaches to treat Shigellosis.

## Materials and methods

### Cell culture
A549 (ATCC #CCL-185), HT-29 (ATCC #HTB-38), and MEFs were cultured in Dulbecco's Modified Eagle Medium (DMEM, Gibco) supplemented with 1% MEM non-essential amino acids (Gibco), 55 µM 2-Mercaptoethanol (Gibco) and 10% heat-inactivated fetal bovine serum (FBS). MEFs were generated from E12 to E14 embryos as previously described (*Coers et al., 2008*). Cells were grown at 37°C in 5% $CO_2$. Cell lines were authenticated using GenePrint10 (Promega) by the Duke University DNA Analysis Facility. All cell lines were routinely tested for mycoplasma.

### Knock out cell lines and mammalian expression systems
The production and characterization of A549 knockout cell lines (pooled RNF213KO, HOIP[KO] clone, and HOIL-1[KO] clone) was previously reported (*Hernandez et al., 2022*; *Walsh et al., 2022*). Various GFP-tagged IpaH expression constructs including the IpaH1.4 and IpaH2.5 C/S mutants were previously described (*de Jong et al., 2016*). Internally Strep-tagged ubiquitin (INT-Ub) and Internally Strep-tagged and N-Terminally HA-tagged ubiquitin (NTerm-HA-Ub_ INT-Strep) open reading frames were cloned via Gateway cloning into the lentiviral expression plasmid pLEX307, a gift from Dr. David Root (Addgene plasmid # 41392; http://n2t.net/addgene:41392; RRID:Addgene_41392). DNA of full-length human RNF213 with N-terminal mCherry was a gift from Daisuke Morito (*Sugihara et al., 2019*). RNF213 was amplified in segments using PCR and cloned using Golden Gate assembly with the enzyme kit BsmBI-v2 (New England Biolabs). Endogenous BsmBI sites were removed from RNF213 by introducing silent mutations in primers used to add enzyme overhangs for Golden Gate assembly. RNF213 was first cloned into a modified Gateway donor vector pDONR221 (Thermo Fisher Scientific) containing a golden gate cloning site in between ATTP sites to make pENTR-mCherry-RNF213. mCherry-RNF213 was transferred into piggybac vector PB-TA-ERN using LR clonase enzyme (Thermo Fisher Scientific) for dox-inducible expression of mCherry-RNF213. PB-TA-ERN (*Kim et al., 2016*) was a gift from Knut Woltjen (Addgene plasmid #80474; http://n2t.net/addgene:80474; RRID:Addgene_80474). Whole plasmid sequencing of PB-TA-ERN-mCherry-RNF213 and the INT-Ub vectors was performed by Plasmidsaurus using Oxford Nanopore Technology with custom analysis and annotation. The INT-Ub protein sequences are provided in *Supplementary file 1*. For mCherry-RNF213 expression in human cells, 293T cells (ATCC CRL-3216) were transfected with PB-TA-ERN and piggybac transposase (pRP[Exp]-EGFP-EF1A>hyPBase, VectorBuilder) using lipofectamine 2000 transfection reagent (Thermo Fisher Scientific), according to the manufacturer's instructions. Cells were grown for approximately 8 days to allow for loss of the transiently transfected vectors, while mCherry-RNF213 that had integrated into the 293T genome was being maintained.

### Production of Rnf213 knockout mouse and derived cells
RNF213 knockout mice were generated using a modified iGONAD methodology (*Skavicus and Heaton, 2023*; *Gurumurthy et al., 2019*; *Ohtsuka et al., 2018*). Briefly, 1 day time-mated, vaginal-plug-positive 8- to 24-week-old C57BL/6J female mice were put under isoflurane anesthesia. A dorsal surgical incision was made, and the ovary/oviduct was exposed. The oviduct was injected with a 0.5-µl Cas9/CRISPR mix which included 6.1 µM Cas9 protein (IDT, Cat. #1081058), 30 µM sgRNAs (IDT, Alt-R CRISPR–Cas9 sgRNA), and 0.02% Fast Green FCF (VWR, Cat. #AAA16520-14) in Opti-MEM (Thermo Fisher Scientific, Cat. #11058021). The 30 µM sgRNA concentration included the following two sgRNAs targeting exon 28 of *Rnf213*: 5'-TTAAATACTGGTAAGGTCGT(TGG)-3' and 5'-AGTCGGAG

TAGCAAAATCCC(TGG)-3'. After the Cas9/sgRNA mix injection, the oviducts were covered with tissue and then electroporated using a CUY21EDIT II (BEX Co, Ltd) electroporator with the settings: Square mA mode, Pd V: 60 V, Pd A: 100 mA, Pd on: 5 ms, Pd off: 50 ms, Pd N: 3, and Decay: 10%. This procedure was done on both sets of oviducts in the mouse. After surgery, female mice were housed together and at 19–21 days post-surgery monitored for pup birth. Pups were genotyped, and three founder mice with out-of-frame 101, 284, or 407 bp deletions within exon 28, respectively, were selected and bred to homozygosity. Loss of Rnf213 protein expression in cells derived from these knockout mice was validated by western blotting.

## Lentivirus production

Plasmids were transfected into HEK 293T cells using the TransIT 293 transfection reagent (Mirus). Wells were transfected with 1 μg plasmid, 750 ng pGAGpol, 250 ng VSVG. Six hours post-transfection, media was aspirated and replaced with fresh DMEM. Twenty-four hours post-transfection, media was aspirated and replaced with pseudoparticle DMEM (3% FBS, 1% MEM non-essential amino acids, 20 mM HEPES). Supernatant was collected 48 and 72 hr after transfection. Supernatants containing lentivirus were filtered using 0.45 μm filters (Corning) and later mixed with 10 μg/ml polybrene (Millipore Sigma). Virus was frozen at –80 until use. For cell transduction, A549 cells were diluted to a concentration of $3.33 \times 10^4$ cells/ml and mixed with 10 μg/ml polybrene. 500 μl of lentivirus was added to each well of a 6-well plate, and 1.5 ml of diluted cells were added on top. Cells were selected in media with 1 μg/ml of puromycin and expanded.

## IpaH mini-screen

HEK 293T that express PiggyBac RNF213 were seeded at $10^5$ cells/well. Sixteen hours before transfection, aTc was added to cells to induce expression of RNF213. The next morning, 2.5 μg of IpaH effector-encoding plasmids were used to transfect HEK 293T using the TransIT 293 transfection reagent (Mirus). Twenty-four hours after transfection, cells were lysed in RIPA buffer and prepared for immunoblotting, as described (*Piro et al., 2017*).

## Bacterial strains

All *S. flexneri* strains were grown on Tryptic Soy Broth (TSB, Millipore Sigma) agar plates in the presence of 0.01% of Congo Red (Millipore Sigma) and red colonies were cultured in TSB broth culture. *S. flexneri* 2457T wild-type and mutant strains (*ΔmxiE and ΔipaH1.4*) were previously described (*Piro et al., 2017*). *S. flexneri* M90T strains (wild-type, *ΔipaH2.5*, *ΔipaH1.4*, and *ΔipaH1.4-ipaH2.5*) were also previously reported (*de Jong et al., 2016*). For immunofluorescence studies, we used fluorescent *S. flexneri* strains carrying the pGFPmut2 (*Cormack et al., 1996*) or the pJUMP45-2A (sfGFP) (*Valenzuela-Ortega and French, 2021*) plasmids. Bacteria were transformed with the ilux pGEX(-) plasmid (Addgene plasmid #107879) (*Gregor et al., 2018*) to generate bioluminescent bacteria, as previously reported (*Dickinson et al., 2023*). Where specified, *S. flexneri* strains were used that harbor the pilT plasmid which encodes an adhesin (*Clerc and Sansonetti, 1987*) and enhances bacterial adhesion to host cells. To complement the *ΔipaH1.4* strain, full-length IpaH1.4 and its catalytically inactive variant C368S were cloned into the bacterial expression vector pDSW206 using Takara In-Fusion Snap Assembly Master Mix (Cat# 638948) following the manufacturer's instructions. Bacterial transformation was performed as previously described (*Piro et al., 2017*). Antibiotics were used as needed at the following concentration: carbenicillin (50 μg/ml), kanamycin (50 μg/ml), spectinomycin (50 μg/ml), and chloramphenicol (100 μg/ml).

## Infection procedures

For immunofluorescence staining, A549, HT29, and MEF cells were seeded at $1 \times 10^5$ cells per well on cover slips in 24-well plates. HT29 cells were plated at $2 \times 10^5$ cells per well. For Western blotting, cells were plated at $2 \times 10^5$ cells per well in 24-well plates. Cells were either left untreated or treated overnight with human or mouse cytokine IFNγ at a final concentration of 100 U/ml, as indicated. The day before infection, *S. flexneri* was inoculated into TSB with antibiotics as needed and grown overnight at 37°C with aeration. The next day, bacteria were diluted (1:20) in TSB with antibiotics and grown for 1 hr 30 min. To induce the expression of WT and C368S mutant IpaH1.4 in complemented *ΔipaH1.4* bacteria, 1 mM IPTG (Invitrogen 15529-019) was added 1 hr after subculturing. Subsequently, the

optical density of the broth culture was measured and pelleted bacteria were washed in sterile PBS and resuspended in DMEM. Infections were carried out at an approximate multiplicity of infection (MOI) of 50–100, unless otherwise specified. Cells were centrifuged at 1000 × *g* for 10 min at the time of infection and infected cells were incubated at 37°C with 5% $CO_2$. Cells were washed with warm HBSS two times after 90 min (A549 and HT29 cells) or 60 min (MEFs) post infection. Following these washes, gentamicin at a final concentration of 50 µg/ml (for A549 and MEFs) or 100 µg/ml (for HT29) was added to the media. For fluorescence microscopy, infected cells were fixed with 4% PFA in phosphate-buffered saline (PBS) at 4 hpi (A549 cells), 3 hpi (HT29), or 2 hpi (MEFs). All *S. flexneri* strains carried the plasmid pGFPmut2 to visualize bacteria by fluorescence microscopy. For western blotting, bioluminescence experiments, and MEF infections, *S. flexneri* strains expressed PilT for improved adhesion and infection rates.

## Immunofluorescence microscopy

Cells were fixed in 4% PFA in PBS and then permeabilized either with 0.1% Triton in PBS for 15 min or with ice-cold methanol for 1 min followed by 0.05% saponin in blocking buffer. Blocking buffer was composed of 5% BSA, 2.2% glycine ± 0.05% saponin or 10% goat serum and 0.05% saponin in PBS. Following permeabilization, coverslips were blocked for 30 min at room temperature (RT). Next, coverslips were incubated with the respective primary antibody diluted in blocking buffer overnight at 4°C. The following primary antibodies were used: rabbit monoclonal anti-RNF213 (Sigma HPA026790; 1:1000), rabbit monoclonal anti-linear (M1) ubiquitin; clone 1E3 (Sigma ZRB2114; 1:250), mouse polyclonal anti-ubiquitin FK2 (Cayman Chemical 14220; 1:100), rabbit monoclonal anti-K63 ubiquitin, clone Apu3 (Sigma 05-1308; 1:100), rabbit monoclonal anti-K27 (Abcam ab181537; 1:100), mouse monoclonal anti-Strep (Genscript A01732; 1:200), custom rabbit anti-mRnf213 antibody from an animal immunized with peptides RKSNEGGNTQPEDQRKPGEGR (aa296–316) and KDTVEYEFI-YEQAQKKGE (aa429–446) (Life Technologies, 1:100). Coverslips were then washed three times with 0.05% Triton in PBS or 0.05% saponin and incubated with an Alexa Fluor-conjugated secondary antibody (1:1000, Invitrogen) and nuclear dye Hoechst 33258 (2 µg/ml; Invitrogen) for 1 hr at RT. Next, coverslips were washed three times with 0.05% Triton or 0.05% saponin in PBS and mounted on glass slides using a mix of mowiol 4-88 combined with para-phenylenediamine antifading agent (9:1). Coverslips were imaged at a Zeiss Axio Imager microscope with Apotome 3 using a 63×/1.4 NA Oil immersion lens. Two to eleven fields of view were captured per coverslip to account for at least 100 bacteria, and z slices with an interval of 0.5 µm were taken per field of view. Images were processed on the software program Fiji. Bacteria that had protein signal surrounding at least 50% of its visible surface were considered as targeted.

## Bacterial luminescence assay

Infections were conducted with bacterial strains carrying the pilT plasmid to promote cell adhesion and the ilux plasmid as a bioluminescence reporter. Host cells were seeded in 96-well plates at a density of 2.5 × 10⁴ cells per well. An approximate MOI of 5 in an infection volume of 100 µl per well was used. Twenty-five minutes post-infection, wells were washed once with 200 µl of warm HBSS and then fresh DMEM containing 50 µg /ml gentamicin was added to each well. Bacterial bioluminescence was measured on a multimode microplate reader (Biotek Synergy H1, Agilent) every 30 min, starting at 1 hpi and finishing at 8 hpi.

## Western blotting

After infection, cells were washed two times with cold HBSS and lysed in RIPA buffer (Sigma) in the presence of 4 U/ml DNAse I (NEB) and protease inhibitor cocktail (Sigma). Lysates were incubated at 4°C for 30 min and centrifuged at 20,000 × *g* for 12 min at 4°C. Protein concentrations were measured using the BCA kit (Thermo Fisher) and normalized. Normalized samples were mixed with Laemmli buffer that contained 5% β-mercaptoethanol and heated at 95°C for 10 min. Samples were heated at 56°C for 10 min when samples were prepared for RNF213 immunoblotting. Ten to thirty µg of protein sample were resolved on a 4–20% Mini-Protean TGX stain-free gels (Bio-Rad) and then blotted onto a PVDF membrane using a tank transfer system overnight at 4°C. After the transfer, PVDF membranes were blocked with 5% nonfat dry milk in Tris-buffered saline containing 0.1% Tween20 (Sigma-Aldrich) (TBS-T) for 1 hr at RT. Subsequently, membranes were incubated with primary antibody diluted in

blocking solution overnight at 4°C. After primary antibody incubation, blots were washed three times with TBST and incubated with the respective secondary antibody in blocking solution for 1 hr at RT. Afterwards, membranes were washed with TBS-T and developed with Clarity ECL (Cytiva). Blots were imaged on an Azure 500 visualization system. Primary antibodies used in this study include rabbit monoclonal anti-RNF213 (Sigma HPA026790; 1:2000), rabbit polyclonal anti-GAPDH (Abcam ab9845; 1:10000); mouse monoclonal anti HOIL-1, clone E3E (Sigma MABC576; 1:500); custom rabbit anti-mRnf213 (Life Technologies, 1:500); rabbit polyclonal anti-HOIP (Abcam ab46322; 1:500); and monoclonal Anti-β-Actin (Sigma A2228; 1:6000). Images were processed in Fiji.

## UBAIT substrate capture assays

UBAIT assays were performed essentially as described before (*Hansen et al., 2021*). Briefly, unknown substrates were captured from colon and cecum lysates from 6- to 8-week-old wild-type C57BL/6 (Charles River) mice. All mouse infection experiments conducted were in accordance with the policies of the Institutional Animal Care and Use Committee at UT Southwestern. Mice were orally infected by gavage with 1x10$^9$ *S. enterica* Typhimurium S1344 WT, and after 4 days mice were euthanized, and colon and cecum were removed, washed, and homogenized in Lysis Buffer (1x ubiquitination buffer (Boston Biochem), 1x protease inhibitor (Sigma), 1 mM DTT). Homogenates were then centrifuged at 13,000 × *g* at 4°C for 10 min and supernatants were added to UBAIT reactions. The supernatant was incubated with 25 µg of purified Strep-IpaH2.5-3xFLAG-Ub or Strep-IpaH3-3xFLAG-Ub and rotated at 4°C. After 1 hr, His-UbE1 (100 nM), His-UbcH5b (2000 nM), energy regeneration buffer (Boston Biochem), MgCl (1 mM), and 1x ubiquitination buffer were added to a final volume of 500 µl. The reaction was run at 30°C for 10 min and then 300 µl TBS buffer (25 mM Tris-HCL pH 7.4, 150 mM NaCl, 1 mM DTT) and 30 µl of washed Strep-Tactin Superflow beads (Iba LifeSciences) were added and rotated for 2 hr at 4°C. The beads were washed four times in TBS-T (TBS + 0.5% Triton-X100) and two times in TBS. Strep-IpaH2.5-3xFLAG-Ub was eluted from the beads in 150 µl Strep-tag elution buffer (Iba LifeSciences). Eluted sample and beads were centrifuged, and supernatant moved to a new tube. SDS (0.25% final concentration) and DTT (5 mM final concentration) were added, and samples were heated at 95°C for 5 min. Next, 1.2 ml TBS was added, followed by 20 µl of M2-FLAG beads (Sigma). The mixture was rotated for 2 hr at 4°C. M2-FLAG beads were washed four times in TBS-T followed by 2 times in TBS. Finally, 35 µl of hot (95°C) SDS–PAGE Loading Buffer was added and the beads were heated for an additional 5 min at 95°C. Samples were electrophoresed on a gradient precast SDS–PAGE gel (Bio-Rad) and Coomassie stained. A lane of gel above the band with unmodified Strep-IpaH2.5-3xFLAG-Ub or Strep-IpaH3-3xFLAG-Ub was excised, and proteins were digested overnight with trypsin (Pierce) following reduction and alkylation with DTT and iodoacetamide (Sigma-Aldrich). Following solid-phase extraction cleanup with an Oasis HLB µelution plate (Waters), the resulting peptides were reconstituted in 10 µl of 2% (vol/vol) acetonitrile (ACN) and 0.1% trifluoroacetic acid in water. Two µl of each sample were injected onto an Orbitrap Fusion Lumos (Thermo) mass spectrometer, coupled to an Ultimate 3000 RSLC-Nano liquid chromatography systems (Thermo) at the Mass Spectrometry Core at University of Texas Southwestern. Samples were injected onto a 75 µm i.d., 75 cm long, (Lumos) EasySpray column (Thermo), and eluted with a gradient from 0 to 28% buffer B over 90 min. Buffer A contained 2% (vol/vol) ACN and 0.1% formic acid in water, and buffer B contained 80% (vol/vol) ACN, 10% (vol/vol) trifluoroethanol, and 0.1% formic acid in water. The Orbitrap Fusion Lumos mass spectrometer operated in positive ion mode with a source voltage of 2.0–2.4 kV and an ion transfer tube temperature of 275°C. MS scans were acquired at 120,000 resolution in the Orbitrap and up to 10 MS/MS spectra were obtained in the Orbitrap for each full spectrum acquired using higher-energy collisional dissociation for ions with charges 2–7. Dynamic exclusion was set for 25 s after an ion was selected for fragmentation. Raw MS data files were analyzed using Proteome Discoverer v.2.4 (Thermo), with peptide identification performed using Sequest HT searching against the human reviewed protein database from UniProt. Fragment and precursor tolerances of 10 ppm and 0.6 Da (Lumos) were specified, and three missed cleavages were allowed. Carbamidomethylation of Cys was set as a fixed modification and oxidation of Met was set as a variable modification. The false-discovery rate cutoff was 1% for all peptides. The abundance of identified peptides found for Strep-IpaH2.5 UBAIT was compared to those found for control IpaH3 UBAIT that was included in the same experiment. Hits for IpaH2.5 were included when abundance levels were at least twofold higher than UBAIT controls.

## Statistical analysis

Graphpad Prism 10.2.2 was used to generate graphs and to perform statistical analyses. Experiments were conducted in three to five biological replicates. Data in graphs are displayed as mean ± SEM. Statistical analyses included Student $t$-test one- and two-way ANOVA followed by Tukey's multiple comparison tests. Significance was considered when p values were <0.05.

## Data storage

A large UBAIT mass spectrometry dataset broadly probing for IpaH protein family interaction partners was originally generated and published in *de Jong et al., 2016*. The numerical values of all other quantified data depicted in data panels in this manuscript are openly available in the Digital Repositories at Duke under the digital object identifier https://doi.org/10.7924/r4vq36v59.

## Acknowledgements

This work was supported by National Institutes of Health grants AI139425 (to JC), AI083359 (to NMA), AI137031 (to NSH), and AI168107 (to NSH); and the Welch Foundation (grant I-1704 to NMA). NSH holds an Investigators in the Pathogenesis of Infectious Disease Award from the Burroughs Wellcome Fund. The funders had no role in study design, data collection, and interpretation, or the decision to submit the work for publication. We would like to thank members of the Coers lab as well as the labs of Drs. Clare Smith, David Tobin, and Edward Miao for providing valuable feedback.

## Additional information

### Funding

| Funder | Grant reference number | Author |
| --- | --- | --- |
| National Institute of Allergy and Infectious Diseases | AI139425 | Jorn Coers |
| National Institute of Allergy and Infectious Diseases | AI083359 | Neal M Alto |
| National Institute of Allergy and Infectious Diseases | AI137031 | Nicholas S Heaton |
| Welch Foundation | I-1704 | Neal M Alto |
| Burroughs Wellcome Fund | | Nicholas S Heaton |
| National Institute of Allergy and Infectious Diseases | AI168107 | Nicholas S Heaton |

The funders had no role in study design, data collection, and interpretation, or the decision to submit the work for publication.

### Author contributions

Luz Saavedra-Sanchez, Conceptualization, Data curation, Formal analysis, Writing – original draft, Writing – review and editing, Investigation, Methodology; Mary S Dickinson, Data curation, Formal analysis, Supervision, Investigation, Methodology; Shruti S Apte, Yifeng Zhang, Formal analysis, Investigation, Methodology; Maarten De Jong, Data curation, Formal analysis; Samantha Skavicus, Methodology; Nicholas S Heaton, Supervision, Funding acquisition, Methodology, Project administration; Neal M Alto, Data curation, Supervision, Funding acquisition, Project administration; Jorn Coers, Conceptualization, Data curation, Formal analysis, Supervision, Funding acquisition, Writing – original draft, Project administration, Writing – review and editing

### Author ORCIDs

Luz Saavedra-Sanchez http://orcid.org/0000-0002-4362-8928
Mary S Dickinson https://orcid.org/0000-0002-7227-9640
Maarten De Jong https://orcid.org/0000-0003-2278-286X
Neal M Alto https://orcid.org/0000-0002-7602-3853

Jorn Coers https://orcid.org/0000-0001-8707-4608

### Ethics

All mouse infection experiments were conducted by the Alto lab at UT Southwestern in accordance with the policies of the Institutional Animal Care and Use Committee at UT Southwestern. All other studies were conducted in the Coers lab at Duke University Medical School and were approved by the Institutional Biosafety Committee (IBC Registration #: 10-6174-01).

Reviewer #1 (Public review): https://doi.org/10.7554/eLife.102714.3.sa1
Reviewer #2 (Public review): https://doi.org/10.7554/eLife.102714.3.sa2
Reviewer #3 (Public review): https://doi.org/10.7554/eLife.102714.3.sa3
Author response https://doi.org/10.7554/eLife.102714.3.sa4

## Additional files

### Supplementary files

MDAR checklist

Supplementary file 1. Ubiquitin constructs.

### Data availability

The numerical values of all other quantified data depicted in data panels in this manuscript are openly available in the Digital Repositories at Duke under the digital object identifier: https://doi.org/10.7924/r4vq36v59.

The following dataset was generated:

| Author(s) | Year | Dataset title | Dataset URL | Database and Identifier |
| --- | --- | --- | --- | --- |
| Saavedra-Sanchez L, Dickinson MS, Apte S, Zhang Y, de Jong M, Skavicus S, Heaton NS, Ito NM, Coers J | 2025 | Data from: The Shigella flexneri effector IpaH1.4 facilitates RNF213 degradation and protects cytosolic bacteria against interferon-induced ubiquitylation | https://doi.org/10.7924/r4vq36v59 | Duke Research Data Repository, 10.7924/r4vq36v59 |

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
