## [Editor Report · eLife Assessment]

In this manuscript, the authors report the **fundamental** finding that a secreted ubiquitin ligase of Shigella, called IpaH1.4, mediates the degradation of a host defense factor, RNF213. The data are **convincing** and represent a major contribution to our understanding of cell-autonomous immunity and bacterial pathogenesis as they provide new mechanistic insight into how the cytosolic bacterial pathogen Shigella flexneri evades IFN-induced host immunity.

---

## [Referee Report · Reviewer #1 (Public review)]

Shigella flexneri is a bacterial pathogen that is an important globally significant cause of diarrhea. Shigella pathogenesis remains poorly understood. In their manuscript, Saavedra-Sanchez et al report their discovery that a secreted E3 ligase effector of Shigella, called IpaH1.4, mediates the degradation of a host E3 ligase called RNF213. RNF213 was previously described to mediate ubiquitylation of intracellular bacteria, an initial step in their targeting to xenophagosomes. Thus, Shigella IpaH1.4 appears to be an important factor to permit evasion of RNF213-mediated host defense. Strengths: The work is focused, convincing, well-performed and important, and the manuscript is well-written. The revised version addressed all the concerns raised during the initial review.

---

## [Referee Report · Reviewer #2 (Public review)]

Summary:

The authors find that the bacterial pathogen Shigella flexneri uses the T3SS effector IpaH1.4 to induce degradation of the IFNg-induced protein RNF213. They show that in the absence of IpaH1.4, cytosolic Shigella is bound by RNF213. Furthermore, RNF213 conjugates linear and lysine-linked ubiquitin to Shigella independently of LUBAC. Intriguingly, they find that Shigella lacking ipaH1.4 or mxiE, which regulates the expression of some T3SS effectors, are not killed even when ubiquitylated by RNF213 and that these mutants are still able to replicate within the cytosol, suggesting that Shigella encodes additional effectors to escape from host defenses mediated by RNF213-driven ubiquitylation.

Strengths:

The authors take a variety of approaches, including host and bacterial genetics, gain-of-function and loss-of-function assays, cell biology, biochemistry, . Overall, the experiments are elegantly designed, rigorous, and convincing.

---

## [Referee Report · Reviewer #3 (Public review)]

Summary:

In this study the authors set out to investigate whether and how Shigella avoids cell autonomous immunity initiated through M1-linked ubiquitin and the immune sensor and E3 ligase RNF213. The key findings are that the Shigella flexneri T3SS effector, IpaH1.4 induces degradation of RNF213. Without IpaH1.4, the bacteria are marked with RNF213 and ubiquitin following stimulation with IFNg. Interestingly, this is not sufficient to initiate the destruction of the bacteria, leading the authors to conclude that Shigella deploys additional virulence factors to avoid this host immune response. The second key finding of this study is that M1 chains decorate the mxiE/ipaH Shigella mutant independent of LUBAC, which is by and large, considered the only enzyme capable of generating M1-linked ubiquitin chains. These findings are fundamental in nature and of general interest.

Strengths and weaknesses:

The data is well-controlled and clearly presented with appropriate methodology. The authors provide compelling evidence that demonstrates that IpaH1.4 is the effector responsible for the degradation of RNF213 via the proteasome and their conclusions are well supported. They have clearly demonstrated how Shigella disarms RNF213-mediated immunity.

This work builds on prior work from the same laboratory that suggests that M1 ubiquitin chains can be formed independently of LUBAC (in the prior publication this related to Chlamydia inclusions). Two key pieces of evidence support this statement - fluorescence microscopy-based images and accompanying quantification in Hoip and Hoil knockout cells for association of M1-ub, using an M1 specific antibody, and the use of an internally tagged Ub-K7R mutant. Whilst it remains possible that the M1 antibody is non-specific, as acknowledged by the authors, the data in supplementary figure 1, comparing K7R-ub and the N-terminally tagged K7R ub variant, provides evidence that during Shigella infection, LUBAC independent M1-ubiquitin chains are indeed formed. This represents an important new angle in ubiquitin biology.

The importance of IFNgamma priming for RNF213 association to the mxiE or ipaH1.4 remains an interesting question that awaits future studies that compare different intracellular bacteria and the role of RNF213.

Overall, the findings are important for the host-pathogen field, cell autonomous/innate immune signaling fields and microbial pathogenesis fields and the work is a very valuable addition to the recent advances in understanding the role of RNF213 in host immune responses to bacteria.

---

## [Author Response]

The following is the authors’ response to the original reviews

**Reviewer #1 (Public review):**
Shigella flexneri is a bacterial pathogen that is an important globally significant cause of diarrhea. Shigella pathogenesis remains poorly understood. In their manuscript, Saavedra-Sanchez et al report their discovery that a secreted E3 ligase effector of Shigella, called IpaH1.4, mediates the degradation of a host E3 ligase called RNF213. RNF213 was previously described to mediate ubiquitylation of intracellular bacteria, an initial step in their targeting of xenophagosomes. Thus, Shigella IpaH1.4 appears to be an important factor in permitting evasion of RNF213-mediated host defense.Strengths:The work is focused, convincing, well-performed, and important. The manuscript is well-written.

We would like to thank the reviewer for their time evaluating our manuscript and the positive assessment of the novelty and importance of our study. We provide a comprehensive response to each of the reviewer’s specific recommendations below and highlight any changes made to the manuscript in response to those recommendations.

**Reviewer #1 (Recommendations for the authors):**
(1) In the abstract (and similarly on p.10), the authors claim to have shown "IpaH1.4 protein as a direct inhibitor of mammalian RNF213". However, they do not show the interaction is direct. This, in my opinion, would require demonstrating an interaction between purified recombinant proteins. I presume that the authors are relying on their UBAIT data to support the direct interaction, but this is a fairly artificial scenario that might be prone to indirect substrates. I would therefore prefer that the 'direct' statement be modified (or better supported with additional data). Similarly, on p.7, the section heading states "S. flexneri virulence factors IpaH1.4 and IpaH2.5 are sufficient to induce RNF213 degradation". The corresponding experiment is to show sufficiency in a 293T cell, but this leaves open the participation of additional 293T-expressed factors. So I would remove "are sufficient to", or alternatively add "...in 293T cells".

We agree with the reviewer and made the recommended changes to the text in the abstract, in the results section on page 7, and in the Discussion on page 11. During the revision of our manuscript two additional studies were published that provide convincing biochemical evidence for the direct interaction between IpaH1.4 and RNF213 (PMID: 40205224; PMID: 40164614). These studies address the reviewer’s concern extensively and are now briefly discussed and cited in our revised MS.

(2) In the abstract the authors state "Linear (M1-) and lysine-linked ubiquitin is conjugated to bacteria by RNF213 independent of the linear ubiquitin chain assembly complex (LUBAC)." However, it is not shown that RNF213 is able to directly perform M1-ubiquitylation. It is shown that RNF213 is required for M1-linked ubiquitylation in IpaH1.4 or MxiE mutants, this is different than showing conjugation is done by RNF213 itself. This should be reworded.

We agree and edited the text accordingly

(3) Introduction: one of the main points of the paper is that RNF213 conjugates linear ubiquitin to the surface of bacteria in a manner independent of the previously characterized linear ubiquitin conjugation (LUBAC) complex. This is indeed an interesting result, but the introduction does not put this discovery in much context. I would suggest adding some discussion of what was known, if anything, about the type of Ub chain formed by RNF213, and specifically whether linear Ub had previously been observed or not.

We now provide context in the Introduction on page 3 and briefly discuss previous work that had implicated LUBAC in the ubiquitylation of cytosolic bacteria. We emphasize that LUBAC specifically generates linear (M1-linked) ubiquitin chains, while the types of ubiquitin linkages deposited on bacteria through RNF213-dependent pathways had remained unidentified.

(4) Figure 3C: is the difference in 7KR-Ub between WT and HOIP KO cells significant? If so, the authors may wish to acknowledge the possibility that HOIP partially contributes to M1-Ub of MxiE mutant Shigella

The frequencies at which bacteria are decorated with 7KR-Ub is not statistically different between WT and HOIP KO cells. We have included this information in the panel description of Figure 3.

(5) On page 11, the authors state that "...we observed that LUBAC is dispensable for M1-linked ubiquitylation of cytosolic S. flexneri ∆ipaH1.4. We found that lysine-less internally tagged ubiquitin or an M1-specific antibody bound to S. flexneri ∆ipaH1.4 in cells lacking LUBAC (HOIL-1KO or HOIPKO) but failed to bind bacteria in RNF213-deficient cells". In fact, what is shown is that M1-ubiquitylation in ∆ipaH1.4 infection is RNF213-dependent (5E), but the work with lysine mutants, HOIP or HOIL-1 KOs are all with ∆mxiE, not ∆ipaH1.4 (3B) in this version of the manuscript. Ideally, the data with ∆ipaH1.4 could be added, but alternatively, the conclusion could be re-worded.

We now include the data demonstrating that staining of ∆ipaH1.4 with an M1-specific antibody is unchanged from WT cells in HOIL-1 KO and HOIP KO cells. These data are shown in supplementary data (Fig. S3E) and referred to on page 9 of the revised manuscript.

(6) The UBAIT experiment should be explained in a bit more detail in the text. The approach is not necessarily familiar to all readers, and the rationale for using Salmonella-infected ceca/colons is not well explained (and seems odd). Some appropriate caution about interpreting these data might also be welcome. Did HOIP or HOIL show up in the UBAIT? This perhaps also deserves some discussion.

As expected, HOIP (listed under its official gene name Rnf31 in the table of Fig.S2B) was identified as a candidate IpaH1.4 interaction partner as the third most abundant hit from the UBAIT screen. Remarkably, Rnf213 was the hit with the highest abundance in the IpaH1.4 UBAIT screen. To address the reviewer’s comments, we now explain the UBAIT approach in more detail and provide the rational for using intestinal protein lysates from *Salmonella* infected mice. The text on page 8 reads as follows: “To investigate potential physical interactions between IpaH1.4 and IpaH2.5, we reanalyzed a previously generated dataset that employed a method known as ubiquitin-activated interaction traps (UBAITs) (32). As shown in Fig. S2A, the human ubiquitin gene was fused to the 3′ end of IpaH2.5, producing a C-terminal IpaH2.5-ubiquitin fusion protein. When incubated with ATP, ubiquitin-activating enzyme E1, and ubiquitin-conjugating enzyme E2, the IpaH2.5-ubiquitin "bait" protein is capable of binding to and ubiquitylating target substrates. This ubiquitylation creates an iso-peptide bond between the IpaH2.5 bait and its substrate, thereby enabling purification via a Strep affinity tag incorporated into the fusion construct (32). IpaH2.5-ubiquitin bait and IpaH3-ubiquitin control proteins were incubated with lysates from murine intestinal tissue. To detect interaction partners in a physiologically relevant setting, we used intestinal lysates derived from mice infected with *Salmonella,* which in contrast to *Shigella* causes pronounced inflammation in WT mice and therefore better simulates human *Shigellosis* in an animal model. Using UBAIT we identified HOIP (Rnf31) as a likely IpaH2.5 binding partner (Fig. S2B), thus confirming previous observations (28) and validating the effectiveness our approach. Strikingly, we identified mouse Rnf213 as the most abundant interaction partner of the IpaH2.5-ubiquitin bait protein (Fig. S2B). Collectively, our data and concurrent reports showing direct interactions between IpaH1.4 and human RNF213 (36, 37) indicate that the virulence factors IpaH1.4 and IpaH2.5 directly bind and degrade mouse as well as human RNF213.”

(7) It would be helpful if the authors discussed their results in the context of the prior work showing IpaH1.4/2.5 mediate the degradation of HOIP. Do the authors see HOIP degradation? If indeed HOIP and RNF213 are both degraded by IpaH1.4 and IpaH2.5, are there conserved domains between RNF213 and HOIP being targeted? Or is only one the direct target? A HOIP-RNF213 interaction has previously been shown (https://doi.org/10.1038/s41467-024-47289-2). Since they interact, is it possible one is degraded indirectly? To help clarify this, a simple experiment would be to test if RNF213 degraded in HOIP KO cells (or vice-versa)?

We appreciate the reviewer’s suggestions. We conducted the proposed experiments and found that WT *S. flexneri* infections result in RNF213 degradation in both WT and HOIP KO cells. Similarly, we found that HOIP degradation was independent of RNF213. We have included these data in Figs. 5A and S3B of our revised submission. A study published during revisions of our paper demonstrates that the LRR of IpaH1.4 binds to the RING domains of both RNF213 and LUBAC (PMID: 40205224). We refer to this work in our revised manuscript.

**Reviewer #2 (Public review):**
Summary:The authors find that the bacterial pathogen Shigella flexneri uses the T3SS effector IpaH1.4 to induce degradation of the IFNg-induced protein RNF213. They show that in the absence of IpaH1.4, cytosolic Shigella is bound by RNF213. Furthermore, RNF213 conjugates linear and lysine-linked ubiquitin to Shigella independently of LUBAC. Intriguingly, they find that Shigella lacking ipaH1.4 or mxiE, which regulates the expression of some T3SS effectors, are not killed even when ubiquitylated by RNF213 and that these mutants are still able to replicate within the cytosol, suggesting that Shigella encodes additional effectors to escape from host defenses mediated by RNF213-driven ubiquitylation.Strengths:The authors take a variety of approaches, including host and bacterial genetics, gain-of-function and loss-of-function assays, cell biology, and biochemistry. Overall, the experiments are elegantly designed, rigorous, and convincing.Weaknesses:The authors find that ipaH1.4 mutant S. flexneri no longer degrades RNF213 and recruits RNF213 to the bacterial surface. The authors should perform genetic complementation of this mutant with WT ipaH1.4 and the catalytically inactive ipaH1.4 to confirm that ipaH1.4 catalytic activity is indeed responsible for the observed phenotype.

We would like to thank the reviewer for their time evaluating our manuscript and the positive assessment of our work, especially its scientific rigor. We conducted the experiment suggested by the reviewer and included the new data in the revised manuscript. As expected, complementation of the ∆ipaH1.4 with WT IpaH1.4 but not with the catalytically dead C338S mutant restored the ability of Shigella to efficiently escape from recognition by RNF213 (Figs. 5C-D).

**Reviewer #2 (Recommendations for the authors):**
The authors should perform genetic complementation of the ipaH1.4 mutant with WT ipaH1.4 and the catalytically inactive ipaH1.4 to confirm that ipaH1.4 catalytic activity is indeed responsible for the observed phenotype.

We performed the suggested experiment and show in Figs. 5C-D that complementation of the ∆ipaH1.4 mutant with WT IpaH1.4 but not with the catalytically dead C338S mutant restored the ability of *Shigella* to efficiently escape from recognition by RNF213. These data demonstrate that the catalytic activity of IpaH1.4 is required for evasion of RNF213 binding to the bacteria.

**Reviewer #3 (Public review):**
Summary:In this study, the authors set out to investigate whether and how Shigella avoids cell-autonomous immunity initiated through M1-linked ubiquitin and the immune sensor and E3 ligase RNF213. The key findings are that the Shigella flexneri T3SS effector, IpaH1.4 induces degradation of RNF213. Without IpaH1.4, the bacteria are marked with RNF213 and ubiquitin following stimulation with IFNg. Interestingly, this is not sufficient to initiate the destruction of the bacteria, leading the authors to conclude that Shigella deploys additional virulence factors to avoid this host immune response. The second key finding of this paper is the suggestion that M1 chains decorate the mxiE/ipaH Shigella mutant independent of LUBAC, which is, by and large, considered the only enzyme capable of generating M1-linked ubiquitin chains.Strengths:The data is for the most part well controlled and clearly presented with appropriate methodology. The authors convincingly demonstrate that IpaH1.4 is the effector responsible for the degradation of RNF213 via the proteasome, although the site of modification is not identified.Weaknesses:(1)The work builds on prior work from the same laboratory that suggests that M1 ubiquitin chains can be formed independently of LUBAC (in the prior publication this related to Chlamydia inclusions). In this study, two pieces of evidence support this statement -fluorescence microscopy-based images and accompanying quantification in Hoip and Hoil knockout cells for association of M1-ub, using an antibody, to Shigella mutants and the use of an internally tagged Ub-K7R mutant, which is unable to be incorporated into ubiquitin chains via its lysine residues. Given that clones of the M1-specific antibody are not always specific for M1 chains, and because it remains formally possible that the Int-K7R Ub can be added to the end of the chain as a chain terminator or as mono-ub, the authors should strengthen these findings relating to the claim that another E3 ligase can generate M1 chains de novo.(2) The main weakness relating to the infection work is that no bacterial protein loading control is assayed in the western blots of infected cells, leaving the reader unable to determine if changes in RNF213 protein levels are the result of the absent bacterial protein (e.g. IpaH1.4) or altered infection levels.(3)The importance of IFNgamma priming for RNF213 association to the mxiE or ipaH1.4 strain could have been investigated further as it is unclear if RNF213 coating is enhanced due to increased protein expression of RNF213 or another factor. This is of interest as IFNgamma priming does not seem to be needed for RNF213 to detect and coat cytosolic Salmonella.Overall, the findings are important for the host-pathogen field, cell-autonomous/innate immune signaling fields, and microbial pathogenesis fields. If further evidence for LUBAC independent M1 ubiquitylation is achieved this would represent a significant finding.

We would like to thank the reviewer for their time evaluating our manuscript and the positive assessment of our work and its significance. We provide a comprehensive response to the main three critiques listed under ‘weaknesses’ and also have responded to each of the reviewer’s specific recommendations below. We highlight any changes made to the manuscript in response to those recommendations.

(1) As the reviewer correctly pointed out, 7KR ubiquitin cannot only be used for linear ubiquitylation but can also function as a donor ubiquitin and can be attached as mono-ubiquitin to a substrate or to an existing ubiquitin chain as a chain terminator. To distinguish between 7KR INT-Ub signals originating from linear versus mono-ubiquitylation, we followed the reviewer’s advice and generated a N-terminally tagged 7KR INT-Ub variant. The N-terminal tag prevents linear ubiquitylation but still allows 7KR INT-Ub to be attached as a mono-ubiquitin. We found that the addition of this N-terminal tag significantly reduced but not completely abolished the number of Δ*mxiE* bacteria decorated with 7KR INT-Ub. These data are shown in a new Fig. S1 and indicate that 7KR lacking the N-terminal tag is attached to bacteria both in the form of linear (M1-linked) ubiquitin and as donor ubiquitin, possibly as a chain terminator. While we cannot rule out that the anti-M1 antibodies used here cross-react with other ubiquitin linkages, we reason that the 7KR data strongly argues that linear ubiquitin is part of the ubiquitin coat encasing IpaH1.4-deficient cytosolic *Shigella.* Collectively, our data show that both linear and lysine-linked (especially K27 and K63) ubiquitin chains are part of the RNF213-dependent ubiquitin coat on the surface of IpaH1.4 mutants. And furthermore, our data strongly indicate that this ubiquitylation of IpaH1.4 mutants is independent of LUBAC.

(2) We used GFP-expressing strains of *S. flexneri* for our infection studies and were therefore able to use GFP expression as a loading control. We have incorporated these data into our revised figures. These new data (Figs. 4A, 5A, and S3B) show that bacterial infection levels were comparable between WT and mutant infections and that therefore the degradation of RNF213 (or HOIP – see new data in Fig. S3B) is not due to differences in infection efficiency.

(3) We agree with the reviewer that the mechanism by which RNF213 binds to bacteria is an important unanswered question. Similarly, whether other ISGs have auxiliary functions in this process or whether binding efficiencies vary between different bacterial species are important questions in the field. However, these questions go far beyond the scope of this study and were therefore not addressed in our revisions.

**Reviewer #3 (Recommendations for the authors):**
(1) An N-terminally tagged K7R-ub should be used as a control to test whether the signal found around the mutant shigella is being added via the N terminal Met into chains. As it is known that certain batches of the M1-specific antibodies are in fact not specific and able to detect other chain types, the authors should test the specificity of the antibody used in this study (eg against different di-Ub linkage types) and include this data in the manuscript.

We agree with the reviewer in principle. The anti-linear ubiquitin (anti-M1) monoclonal antibody, clone 1E3, prominently used in this study was tested by the manufacturer (Sigma) by Western blotting analysis and according to the manufacturer “this antibody detected ubiquitin in linear Ub, but not Ub K11, Ub K48, Ub K63.” However, this analysis did not include all possible Ub linkage types and thus the reviewer is correct that the anti-M1 antibody could theoretically also detect some other linkage types. To address this concern, we added new data during revisions demonstrating that 7KR INT-Ub targeting to *S. flexneri* is largely dependent on the N-terminus (M1) of ubiquitin. Our combined observations therefore overwhelmingly support the conclusion that linear (M1-linked) as well as K-linked ubiquitin is being attached to the surface of IpH1.4 *S. flexneri* bacteria in an RNF213-dependent and LUBAC-independent manner.

(2) The M1 signal detected on bacteria with the antibody is still present in either Hoip or Hoil KO’s but due to the potential non-specificity of the antibody, the authors should test whether K7R ub is detected on bacteria in the Hoil ko (in addition to Hoip KO). This would strengthen the authors’ data on LUBAC-independent M1 and is important because Hoil can catalyse non-canonical ubiquitylation.

The specific linear ubiquitin-ligating activity of LUBAC is enacted by HOIP. We show that linear ubiquitylation of susceptible *S. flexneri* mutants as assessed by anti-M1 ubiquitin staining or 7KR INT-Ub recruitment occurs in HOIPKO cells at WT levels (Figs. 3B, 3C, S3E [new data]). In our view , these data unequivocally show that the observed linear ubiquitylation of cytosolic *S. flexneri ipaH1.4* and *mxiE* mutants is independent of LUBAC.

(3) For Figure 4A, do mxiE bacteria show similar invasion - authors should include a bacterial protein control to show levels of bacteria in WT and mxiE infected conditions. A similar control should be included in Figure 5A.

We used GFP-expressing strains of *S. flexneri* for our infection studies and were therefore able to use GFP expression as a loading control. We have incorporated these data into our revised figures. These new data (Figs. 4A, 5A, and S3B) show that bacterial infection levels were comparable between WT and mutant infections and that therefore the degradation of RNF213 (or HOIP – see new data in Fig. S3B) is not due to differences in infection efficiency.

(4) Can the authors speculate why IFNg priming is needed for the coating of Shigella mxiE mutant but not in the case of Salmonella or Burkholderia? Is this just amounts of RNF213 or something else?

In our studies we did not directly compare ubiquitylation rates of cytosolic *Shigella, Burkholderia,* and *Salmonella* bacteria with each other under the same experimental conditions. However, such a direct comparison is needed to determine whether IFNgamma priming is required for RNF213-dependent bacterial ubiquitylation of some but not other pathogens. Two papers published during the revisions of our manuscript (PMID: 40164614, PMID: 40205224) reports robust RNF213 targeting to IpaH1.4 *Shigella* mutants in unprimed cells HeLa cells (whereas we used A549 and HT29 cells). Therefore, differences in reagents, cell lines, and/or other experimental conditions may determine whether IFNgamma priming is necessary to observe substantial RNF213 translocation to cytosolic bacteria.

(5) Typos - there are several, but this is hard to annotate with line numbers so the authors should proofread again carefully.

We proofread the manuscript and corrected the small number of typos we identified